# Oligogenic heterozygous inheritance of sperm abnormalities in mouse

Guillaume Martinez[1,2]*, Charles Coutton[1,2], Corinne Loeuillet[1], Caroline Cazin[1,3], Jana Muroňová[1], Magalie Boguenet[1], Emeline Lambert[1], Magali Dhellemmes[1], Geneviève Chevalier[1], Jean-Pascal Hograindleur[1], Charline Vilpreux[1], Yasmine Neirijnck[4], Zine-Eddine Kherraf[1,3], Jessica Escoffier[1], Serge Nef[4], Pierre F Ray[1,3], Christophe Arnoult[1,5]*

[1]Institute for Advanced Biosciences, INSERM, CNRS, Université Grenoble Alpes, Grenoble, France; [2]UM de Génétique Chromosomique, Hôpital Couple-Enfant, CHU Grenoble Alpes, Grenoble, France; [3]UM GI-DPI, CHU Grenoble Alpes, Grenoble, France; [4]Department of Genetic Medicine and Development, University of Geneva Medical School, Genève, Switzerland; [5]Station de Primatologie, UPS 846, CNRS, Rousset, France

*For correspondence:
gmartinez@chu-grenoble.fr
(GM);
christophe.arnoult@univ-grenoble-alpes.fr (CA)

**Competing interest:** The authors declare that no competing interests exist.

**Abstract** Male infertility is an important health concern that is expected to have a major genetic etiology. Although high-throughput sequencing has linked gene defects to more than 50% of rare and severe sperm anomalies, less than 20% of common and moderate forms are explained. We hypothesized that this low success rate could at least be partly due to oligogenic defects – the accumulation of several rare heterozygous variants in distinct, but functionally connected, genes. Here, we compared fertility and sperm parameters in male mice harboring one to four heterozygous truncating mutations of genes linked to multiple morphological anomalies of the flagellum (MMAF) syndrome. Results indicated progressively deteriorating sperm morphology and motility with increasing numbers of heterozygous mutations. This first evidence of oligogenic inheritance in failed spermatogenesis strongly suggests that oligogenic heterozygosity could explain a significant proportion of asthenoteratozoospermia cases. The findings presented pave the way to further studies in mice and man.

## Editor's evaluation

This study provides insights into the detrimental effect of accumulative heterozygous mutations on sperm abnormalities. By breeding a series of knockout strains known to cause multiple flagellar defects, the authors demonstrated that such variations at two (digenic) or more loci (oligogenic) can contribute to sperm abnormalities in the head. These findings are significant in that they implicate oligogenic inheritance as a possible cause of unexplained male infertility.

## Introduction

Infertility is a major health concern affecting 15% of couples of reproductive-age worldwide (*Mascarenhas et al., 2012*; *Boivin et al., 2007*). The infertility burden has increased globally for both genders in the past 30 years (*Sun et al., 2019*), and now affects approximately 50 million couples worldwide (*Datta et al., 2016*). Infertility is broadly treated by assisted reproductive technologies (ART), and today the number of individuals who were conceived by ART is close to 0.1% of the total world population, with over 8 million children already born following in vitro fertilization (IVF). Currently, ART is estimated to account for 1–6% of births in most countries (*Faddy et al., 2018*), with over 2.5 million

cycles performed every year, resulting in over 500,000 births worldwide annually (*Fauser, 2019*). Despite this undeniable success, almost half the couples who seeks medical assistance for infertility fail to achieve successful pregnancy, and nearly 40% of infertile couples worldwide are simply diagnosed with unexplained or idiopathic infertility (*Sadeghi, 2015*). In the clinical context, few efforts are currently being made to understand and specifically address the underlying causes of a couple's infertility because ART can often successfully rescue fertility even without a molecular diagnosis. Nevertheless, this absence of identification of the causes of infertility means that we lack alternative treatments for couples for whom current therapies are unsuccessful. Consequently, it is essential to improve the molecular diagnosis of infertility.

Human male infertility is a clinically heterogeneous condition with a complex etiology, in which genetic defects play a significant role. It is estimated that half of idiopathic cases of male infertility could be attributed to an as-yet unidentified genetic defect (*Krausz, 2011*). However, characterization of the molecular causes of male infertility represents a significant challenge, as over 4000 genes are thought to be involved in sperm production (*Jan et al., 2017*). Over the past decade, significant progress has been made in gene identification thanks to the emergence of next-generation sequencing (NGS), and in functional gene validation thanks to new gene-editing techniques such as CRISPR/Cas9. NGS provides an inexpensive and rapid genetic approach through which to discover novel disease-associated genes (*Fernandez-Marmiesse et al., 2018*). It has proven to be a highly powerful tool in the research and diagnosis context of male infertility (*Xavier et al., 2021*; *Krausz and Riera-Escamilla, 2018*). In addition, validation of newly-identified variants through functional experiments has greatly benefited from the ability to generate mouse knock-out models using CRISPR technology (*Kherraf et al., 2018*) and the use of novel model organisms like *Trypanosoma brucei* to study specific phenotypes, such as multiple morphological abnormalities of the [sperm] flagella (MMAF) syndrome (*Coutton et al., 2018*; *Lorès et al., 2019*; *Martinez et al., 2020*). These developments have resulted in a diagnostic yield, based on known genetic causes, explaining about 50% of cases of rare qualitative sperm defects like globozoospermia, acephalic, or MMAF syndromes (*Beurois et al., 2020*; *Touré et al., 2021*). In contrast, the diagnostic yield for quantitative sperm abnormalities such as oligozoospermia or azoospermia remains below 20%, even though these are the most common forms of male infertility (*Krausz, 2011*; *Krausz and Riera-Escamilla, 2018*; *Tüttelmann et al., 2018*). To improve these low diagnostic yields, international consortia have been created to attempt to identify very low-frequency variants (https://gemini.conradlab.org/ and https://www.imigc.org/). In addition, several groups have started to assemble cohorts of patient-parent trios, aiming to identify de novo mutations causing male infertility as well as providing insight into dominant maternal inheritance (*Xavier et al., 2021*; *Veltman and Brunner, 2012*; *McRae, 2017*). Phenotypic heterogeneity and apparent incomplete penetrance were observed for some genetic alterations involved in male infertility (*Vogt, 2005*; *Röpke et al., 2013*; *Kherraf et al., 2017*). These observations are difficult to reconcile with a model of Mendelian inheritance. We thus raised the possibility that the low diagnostic yield is partly due to the complex etiology of infertility, and hypothesized that some of the unsolved cases are due to oligogenic events, that is, accumulation of several rare hypomorphic variants in distinct, functionally connected genes, and in particular to oligogenic heterozygosity. The molecular basis of oligogenicity is poorly understood. The main hypotheses are that two or more mutant proteins may act at different levels in the same intracellular pathway and could quantitatively contribute to its progressive dysfunction. When a critical threshold is reached, the disease phenotype would emerge. Alternatively, the mutant nonfunctional proteins produced may be part of the same multiprotein complex, and the presence of numerous pathogenic variants would increase the chance of the complex becoming compromised – leading to a progressive collapse of its cellular function (*Kousi and Katsanis, 2015*).

Here, we addressed the oligogenic heterozygosity hypothesis in male infertility using four specially-generated MMAF knock-out (KO) mouse models with autosomal recessive inheritance (*Coutton et al., 2018*; *Coutton et al., 2019*). Following extensive cross-breeding of our KO mouse lines, we produced lines harboring between one and four heterozygous truncating mutations. We assessed and compared fertility for these lines, analyzing both quantitative and qualitative sperm parameters, and performed a fine analysis of sperm nuclear morphology for all strains. Using this strategy, we were able to describe new genetic inheritance of sperm deficiencies.

## Results

### Selection and characterization of individual MMAF mouse lines

To generate mice carrying up to four heterozygous truncating mutations, we first selected four lines carrying mutations in genes inducing a MMAF phenotype, namely *Cfap43*, *Cfap44*, *Armc2*, and *Ccdc146*. Three of these lines were already available and previously reported by our laboratory: a strain with a 4 bp deletion in exon 21 (delAAGG) for Cfap43 (*Coutton et al., 2018*), a strain with a 7 bp insertion in exon 3 (InsTCAGATA) for *Cfap44* (*Coutton et al., 2018*), and a strain with a one-nucleotide duplication in exon 4 (DupT) for Armc2 (*Coutton et al., 2019*), inducing a translational frameshift that leads to the production of a truncated protein. We generated the fourth strain using CRISPR-Cas9 technology as described in the Materials and methods section, inducing a 4 bp deletion in exon 2 (delTTCG) of the *Ccdc146* gene (*Appendix 1—figure 1A*). A study describing how this mutation in the *Ccdc146* gene affects spermatogenesis is currently under review elsewhere (for the reviewers only, we provide the complete phenotype for the Ccdc146 KO mouse strain, demonstrating its role in MMAF syndrome in mice).

We first confirmed the MMAF phenotypes for all four strains. Sperm from all homozygous KO male mice displayed more than 95% morphological abnormalities of the flagellum including coiled, bent, irregular, short or/and absent flagella (*Figures 1–4*). We then analyzed sperm morphology (head and flagellum) in heterozygous animals by optical microscopy. The four strains fell into two categories. For the two strains targeting *Cfap43* and *Cfap44*, *Cfap43*$^{+/-}$, or *Cfap44*$^{+/-}$ +/- males had higher rates ( + 4% and + 12.75%) of abnormalities than wild-type mice (*Figures 1A and 2A*; *Cfap43*: t = −2.79, df = 6.13, p-value = 0.03; *Cfap44*: t = −8.80, df = 6.14, p-value = 0.0001). These results were in accordance with previous observations (*Coutton et al., 2018*). In contrast, for strains harboring heterozygous mutations in *Armc2* and *Ccdc146*, no significant differences were observed with respect to wild-type males (*Figures 3A and 4A*).

Because head morphology defects may be subtle and difficult to detect by visual observation, we applied a newly-developed method, involving the use of Nuclear Morphology Analysis Software (NMAS) (*Skinner et al., 2019*). The method is described in detail in the Materials and methods section. The nuclear morphologies of sperm from each genotype (WT, heterozygotes, and homozygotes) were characterized. For all corresponding wild-type strains, shape modeling gave extremely similar consensus and angle profiles for nuclei, highlighting the common genetic background of the KO animals (*Appendix 1—figure 2*). For heterozygous *Cfap43*$^{+/-}$, *Cfap44*$^{+/-}$, and *Ccdc146*$^{+/-}$ +/-, the angle profiles were very similar to WT profiles (*Figures 1B, 2B and 4B*). In contrast, *Armc2*$^{+/-}$ +/- showed a slightly modified angle profile (–6° at position 365 and +6° at position 435) compared to the WT profile, due to a narrower caudal base inducing a more pronounced caudal bulge and a reduced dorsal angle (*Figure 3B*). When the profiles for all heterozygous males were compared, a very similar nuclear morphology with nearly identical angle and variability profiles was found, apart from for *Armc2*$^{+/-}$, which had slightly more variability (IQR *Armc2*$^{+/-}$ +/- than other curves with +10 IQR in position 350 and +5 IQR in position 450) than the other lines despite a similar angle profile (*Appendix 1—figure 3*).

We also compared these profiles with those of the corresponding KO mice, which displayed very unique and specific patterns (*Figures 1B–4B*). Briefly, *Cfap43*$^{-/-}$ and *Cfap44*$^{-/-}$ angle profiles both displayed a flattening from the ventral angle to the dorsal angle leading to their characteristic 'pepper' shape. The lines for both profiles merged over the majority of regions such as the under-hook concavity and the acrosomal curve, and showed intermediate impairment, greater than that observed in the *Armc2*$^{-/-}$ line and less than that observed in the *Ccdc146*$^{-/-}$ line. The angle profile for *Armc2*$^{-/-}$ mice was similar to that of wild-type mice at the tip, dorsal angle, and acrosomal curve, as these regions do not appear to be affected by the mutation. Under-hook concavity, ventral angle, and tail socket regions were slightly affected, although less than in other lineages. In contrast to the other lineages, the caudal base was markedly shortened, with a significant flattening observed. For the *Ccdc146*$^{-/-}$ line, with the exception of the ventral-vertical region, which was unaffected in any lineage, all regions of the angle profile were strongly impacted. The dorsal angle was completely absorbed into the acrosomal curve, and the under-hook concavity, ventral angle and caudal bulge regions were flattened. The tail socket region even displayed an inverted angular profile compared to the other lineages. Overall, the profile of *Ccdc146*$^{-/-}$ showed considerable variability, and complete remodeling of the angle profile with, among other changes, total inversion of the curve at position 330–400, corresponding to complete disappearance of the tail socket. Comparison of the profiles for KO

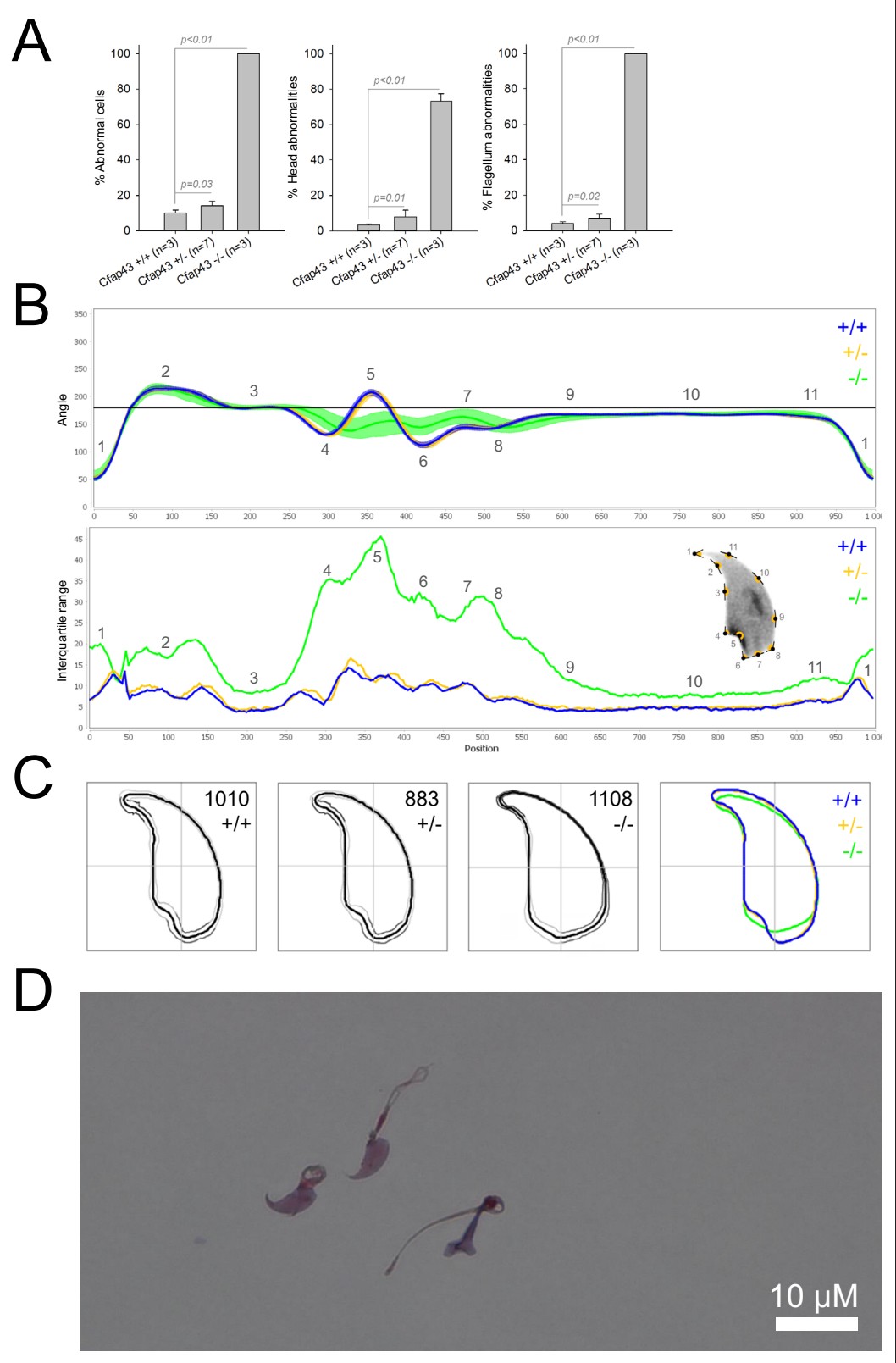

**Figure 1.** Sperm morphology analysis for the *Cfap43* mouse strain.
 (**A**) Histogram showing proportions of morphological anomalies (mean ± SD) for each *Cfap43* genotype observed by light microscopy. Statistical significance was assessed by applying an unpaired Welch *t*-test; p-values are indicated. (**B**) Angle profiles (top) and variability profiles (bottom) from *Cfap43*[+/+] (blue), *Cfap43*[+/-] (yellow), and

*Figure 1 continued on next page*

*Figure 1 continued*

*Cfap43*-/- (green) mice. The x axis represents an index of the percentage of the total perimeter as measured counterclockwise from the apex of the sperm hook. The y axis, corresponding to the angle profile, represents the interior angle measured across a sliding window centered on each index location. The y axis, corresponding to the variability profile, represents the Interquartile Range (IQR) value (the difference between the 75th and 25th percentiles). Specific regions of the nuclei are mapped on the profile and the graphical representation (from *Skinner et al., 2019*) with: 1-tip; 2-under-hook concavity; 3-vertical; 4-ventral angle; 5-tail socket; 6-caudal bulge; 7-caudal base; 8-dorsal angle; 9–11-acrosomal curve. (**C**) Consensus nuclear outlines for each genotype alongside a merged consensus nucleus (blue = *Cfap43*+/+, yellow = *Cfap43*+/-, green = *Cfap43*-/-). The numbers assigned to each consensus outline correspond to the number of nuclei processed per condition. (**D**) Optical microscopy analysis showing a representative MMAF phenotype for *Cfap43* KO mice (scale bar, 10 μm).

sperm (*Appendix 1—figure 4*) showed that *Cfap43*-/- and *Cfap44*-/- mice displayed similar alterations (general enlargement of the head, rounding of the base at the expense of the flagellum insertion, etc.) resulting in close consensus; the other two lines were quite distinct. Although it was the most affected heterozygous line, *Armc2*-/- mice displayed the mildest alterations to nuclear morphology and variability of the four lines. KO animals nonetheless retained specific patterns, including modification of the hook curve and a specific slope of the base profile. Finally, *Ccdc146*-/- mice displayed extremely severe alterations, with almost all of their nuclei presenting a triangular shape (*Figure 4C–D*). In line with previous studies (*Coutton et al., 2018*; *Lorès et al., 2019*; *Coutton et al., 2019*; *Hwang et al., 2021*), these results demonstrate that absence of MMAF genes not only affects sperm flagellum biogenesis but can also have an impact on sperm head morphology.

From these extensively characterized lines, we then proceeded to generate mice harboring multiple mutations.

## Impact of accumulation of heterozygous mutations

All mice harboring between one and four heterozygous truncating mutations were generated by standard cross-breeding of the four lines. A considerable number of generations were produced over several years. Due to time and financial constraints, only one combination of multiple heterozygous lines was created and analyzed. Double heterozygotes were obtained by crossing *Cfap43* and *Cfap44* KO mouse lines. Triple heterozygous animals were mutated for *Cfap43*, *Cfap44*, and *Armc2*; and the quadruple heterozygous line also had the *Ccdc146* mutation (*Figure 5*).

The accumulation of heterozygous mutations on the four selected genes involved in MMAF syndromes induced qualitative spermatogenesis defects (illustrated in *Appendix 1—figure 5*), with progressive increased numbers of morphological anomalies (from 9.81% ± 2.48% for control mice to 42.33% ± 3.78% for mice bearing four mutations, a 430% increase. *Figure 5A*), in particular defects of the head (from 6.56% ± 2.30% for control mice to 40.33% ± 3.78% for mice bearing four mutations, a 615% increase. *Figure 5B*). Males harboring one mutation exhibited a significant increase in flagellar abnormalities (*Figure 5C*; see also 6 C). However, flagellar anomalies were not amplified as the number of mutated genes increased (t = −2.78, df = 2.41, p-value = 0.08, for control versus four mutations. *Figure 5C*). Nevertheless, accumulation of mutations had a negative impact on sperm motility parameters (illustrated in *Videos 1–5*). Although the decrease of the overall percentage of motile cells is not significant with increasing numbers of mutations (from 38.55 ± 11.50 μm/s for control to 29.13 ± 6.02 μm/s for mice bearing four mutations, t = 1.48, df = 4.78, p-value = 0.20. *Figure 5D*), the quality of sperm movement was strongly affected (*Figure 5E–F*). Thus, for sperm bearing two mutations, the average sperm velocity and straight-line velocity were halved (−44.6% and −46.1% respectively) compared to control mice (VAP: 92.31 ± 23.38–51.16 ± 16.59 μm/s,; VSL: 81.52 ± 20.84–43.96 ± 13.75 μm/s), and the decreasing trend continued as mutations accumulated (41.18 ± 10.27 and 33.43 ± 8.62 μm/s for VAP and VSL of mice bearing four mutations, that is −55.4% and −59% versus control mice). It is worth noting that the deterioration of the morphological phenotype with the addition of new heterozygous mutations occurs even though the *Ccdc146* and *Armc2* heterozygous mutations alone had no impact on the sperm phenotype observed by optical microscopy (*Figures 3A and 4A*).

From the crosses performed to generate multi-heterozygous animals, other combinations of heterozygous mutations were obtained. The sperm parameters of the corresponding animals were also

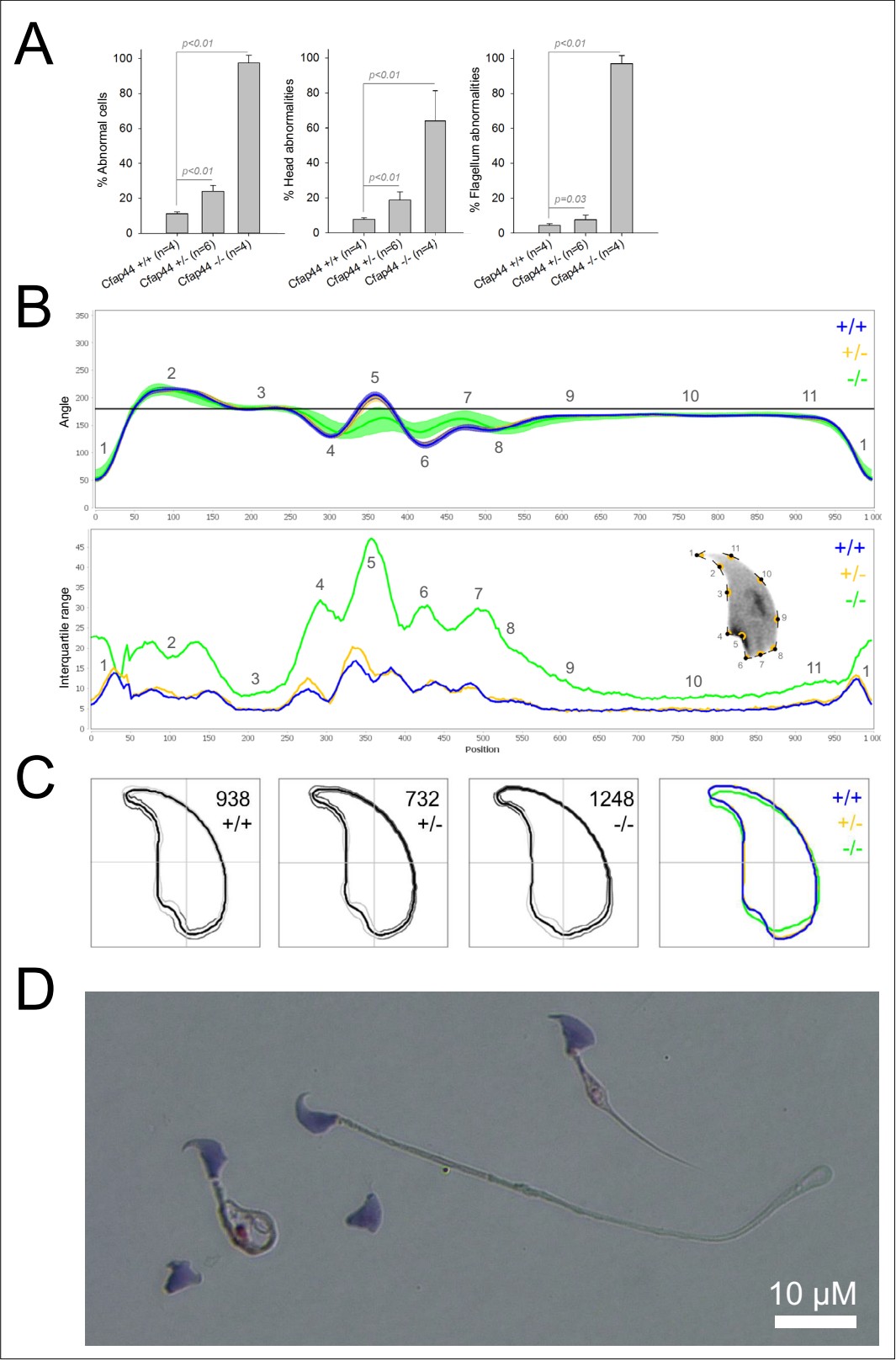

**Figure 2.** Sperm morphology analysis for the *Cfap44* mouse strain.
(**A**) Histogram showing proportions of morphological anomalies (mean ± SD) for each *Cfap44* genotype. Statistical significance was assessed by applying an unpaired Welch *t*-test; p-values are indicated. (**B**) Angle profiles (top) and variability profiles (bottom) for *Cfap44*^+/+^ (blue), *Cfap44*^+/-^ (yellow), and *Cfap44*^-/-^ (green) mice. The x axis represents

*Figure 2 continued on next page*

*Figure 2 continued*

an index of the percentage of the total perimeter, as measured counterclockwise from the apex of the sperm hook. The y axis corresponding to the angle profile is the interior angle measured across a sliding window centered on each index location. The y axis corresponding to the variability profile represents the Interquartile Range (IQR) value (the difference between the 75th and 25th percentiles). Specific regions of the nuclei are mapped on the profile and the graphical representation (from *Skinner et al., 2019*) with: 1-tip; 2-under-hook concavity; 3-vertical; 4-ventral angle; 5-tail socket; 6-caudal bulge; 7-caudal base; 8-dorsal angle; 9–11-acrosomal curve. (**C**) Consensus nuclear outlines for each genotype alongside a merged consensus nucleus (blue = *Cfap44⁺/⁺*, yellow = *Cfap44⁺/⁻*, green = *Cfap44⁻/⁻*). The numbers assigned to each consensus outline correspond to the number of nuclei processed per condition. (**D**) Optical microscopy analysis showing a representative MMAF phenotype for *Cfap44* KO mice (scale bar, 10 μm).

phenotyped. Interestingly, rates of morphological defect (*Figure 6A–C*, red dots) and motility parameters (*Figure 6D–F*, red dots) were very similar whatever the gene combinations. Taken together, these results show that it is not the specific combination of heterozygous mutations that leads to altered sperm morphology and sperm motility parameters, but rather their accumulation.

Despite a marked alteration of spermatocytograms, accumulation of mutations did not have a significant effect on quantitative spermatogenesis defects. For example, sperm production – represented by testis weight and sperm concentration – and overall fertility of the animals – based on the number of pups per litter and interval between litters – were not affected, whatever the combination and number of mutations (testis weight: t = −0.80, df = 2.33, p-value = 0.49; sperm concentration: t = −0.89, df = 5.68, p-value = 0.40; time/litter: t = 0.70, df = 25.98, p-value = 0.48; pups/litter: t = 0.80, df = 34.68, p-value = 0.42; for control versus mice bearing four mutations. *Figure 7*).

As for mouse lines bearing single mutations, we then used NMAS to help characterize head anomalies in multi-heterozygote mutated lines. Our results indicated a striking progressive deterioration in sperm head morphology as mutations accumulated (*Figure 8*). Each additional mutation progressively and significantly increased the variability and severity of the morphological defects observed in the nucleus, negatively influencing angle and consensus profiles (perimeter and max feret progressively decline from 49.12 ± 0.07 and 19.01 ± 0.03 μm to 44.23 ± 0.21 and 16.41 ± 0.11 μm between mice bearing one and four mutations for example). This negative effect was cumulative, and if a threshold or plateau effect exists, it was not reached upon accumulation of four mutations.

To extend our analysis, we then used the software to analyze sperm sub-populations by performing unbiased nuclear morphology categorization. Clustering based on angle profiles revealed a total of ten sub-groups of nuclei shape, which matched with the usual shapes of normal and abnormal mouse sperm, defined more than 30 years ago (*Krzanowska, 1981*). Mutation-accumulation progressively increased the frequency of all abnormal shapes and decreased the frequency of normal forms. The most unstructured forms were associated with the highest number of mutations (*Appendix 1—figure 6*).

## Discussion

The aim of this study was to determine whether the accumulation of several rare heterozygous variants in functionally connected genes affected fertility and sperm parameters in mice. Our results clearly demonstrated that spermatogenesis failure can arise from oligogenic heterozygosity in mice. Males bearing increased numbers of heterozygote mutations in genes involved in MMAF syndrome exhibited altered spermatocytogram, with significant increase of proportion of abnormal sperm and decreased sperm motility. Both defective sperm motility and abnormal sperm head morphology could negatively impact fertility. Concerning sperm motility, it has been clearly shown that the percentage of conception depends on the total number of motile sperm within the semen (*Publicover and Barratt, 2011*). Sperm motility is indeed a key parameter for fertilization, which occurs deeply in the mammalian female tract. Motility is necessary not only for the sperm to reach the oocyte in the oviduct, but sperm motility is also required for the sperm to cross the protective layers of the oocytes and is essential for gamete fusion (*Ravaux et al., 2016*). Concerning abnormal head morphology, there are numerous reports of a significant correlation between morphology and infertility and it is accepted that any increase in any sperm abnormality should be regarded as a possible cause of decreased fertility, and that precise analyses of sperm abnormalities is a useful approach for diagnosis or research

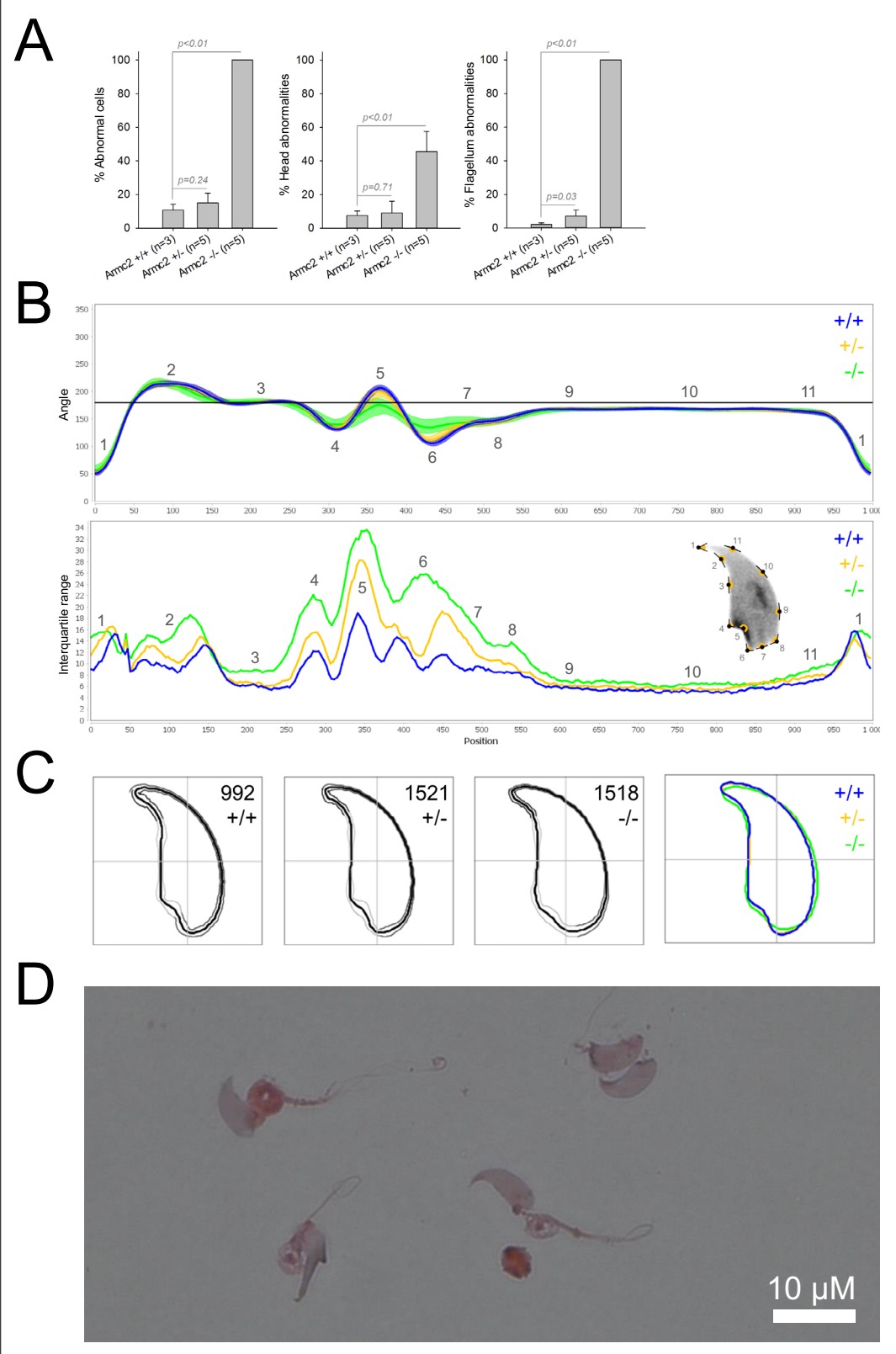

**Figure 3.** Sperm morphology analysis for the *Armc2* mouse strain.
 (**A**) Histogram showing proportions of morphological anomalies (mean ± SD) for each *Armc2* genotype. Statistical significance was assessed by applying an unpaired Welch *t*-test; p-values are indicated. (**B**) Angle profiles (top) and variability profiles (bottom) for *Armc2*^+/+^ (blue), *Armc2*^+/-^ (yellow), and *Armc2*^-/-^ (green) mice. The x axis represents

*Figure 3 continued*

an index of the percentage of the total perimeter, as measured counterclockwise from the apex of the sperm hook. The y axis corresponding to the angle profile represents the interior angle measured across a sliding window centered on each index location. The y axis corresponding to the variability profile represents the Interquartile Range (IQR) value (the difference between the 75th and 25th percentiles). Specific regions of the nuclei are mapped on the profile and the graphical representation (from *Skinner et al., 2019*) with: 1-tip; 2-under-hook concavity; 3-vertical; 4-ventral angle; 5-tail socket; 6-caudal bulge; 7-caudal base; 8-dorsal angle; 9–11-acrosomal curve. (**C**) Consensus nuclear outlines for each genotype alongside a merged consensus nucleus (blue = *Armc2$^{+/+}$*, yellow = *Armc2$^{+/-}$*, green = *Armc2$^{-/-}$*). The numbers assigned to each consensus outline correspond to the number of nuclei processed per condition. (**D**) Optical microscopy analysis showing a representative MMAF phenotype for *Armc2* KO mice (scale bar, 10 µm).

purposes (*Andrade-Rocha, 2001*; *Chemes and Rawe, 2003*; *Menkveld et al., 2011*; *Auger et al., 2016*). These findings are supported by the World Health Organization which concludes in its report that 'categorizing all abnormal forms of spermatozoa may be of diagnostic or research benefit'.

However, despite significant sperm parameter alterations, we did not observe here effect on male fertility, suggesting that the threshold leading to the partial/complete collapse of the male reproductive function was not reached. This concept of threshold is strongly associated with oligogenicity. It is defined by the number of mutations within the same multiprotein complex or intracellular pathway beyond which a disease phenotype will be observed (*Dipple and McCabe, 2000*). In this study, we induced mutations in four genes coding for proteins participating in flagella formation and function. Although we did observe a negative cumulative burden, we did not reach the complex or system threshold that would lead to the emergence of a dichotomous severe infertility phenotype. This may be related to the fact that the mouse model has proved limitations to decipher the function of the proteins involved in sperm physiology and to assess the impact of their lack on fertilization. These limitations are the housing conditions, the mating protocol and the very high fertility of this model. For instance, concerning the housing conditions, the function of MAGE cancer testis antigens to protect the male germline is revealed only when males are subjected to an environmental stress (*Fon Tacer et al., 2019*). For mating protocol, the phenotyping is performed in particular conditions where mutated males are mated with wild-type females, without competition and in breeding conditions masking complex phenotypes. An example of such a limitation of classical reproductive phenotyping has been emphasized in the study on the importance of the Pkdrej protein in sperm capacitation (*Sutton et al., 2008*). Another remarkable exemple is the phenotype of Enkurin KO mice, with variable impact on litter size, despite the function of the protein in flagellum beating (*Jungnickel et al., 2018*). Overall, the highly significant increase of sperm defects (decrease of sperm motility and increase of head morphological defects) induced by the accumulation of deficient proteins shown in this study would probably impact male fertility in more challenging conditions in mouse. Moreover, the mouse model has much higher fertility than human. Human spermatogenesis is clearly less efficient than that of mice and the level of male infertility is around 15% whereas is less than 1% in mice. It is therefore to be expected that the various defects we noticed in this study will impact more severely human sperm fertilizing competence.

A similar mutational burden in humans, known to have the highest number of sperm defect among primates (*Martinez and Garcia, 2020*), could, however, be sufficient to reach the threshold for fertility collapse. It is worth noting that the probability of accumulating this type of heterozygous mutations in testis is considerable, because (i) thousands of genes are necessary to achieve spermatogenesis (*Jan et al., 2017*), (ii) expression of most spermatogenesis-associated genes is restricted to or strongly enriched in the testis (*Uhlén et al., 2016*), and consequently (iii) the risk of life-threatening impact of mutations is limited.

Moreover, mutant gene products retaining some residual function could be influenced by additional systemic perturbation (see review in *Vander Borght and Wyns, 2018*) that would lead to system collapse. For instance, environmental factors commonly responsible for milder alterations to spermatogenesis could play an important role by severely aggravating the genetic burden on the system. This type of multi-factorial input could explain the phenotypic continuum observed in patients with idiopathic infertility.

In this article, we showed that the haplo-insufficiency of several genes involved in flagellum biogenesis and associated with MMAF syndrome leads to head defects. When one thinks of MMAF patients,

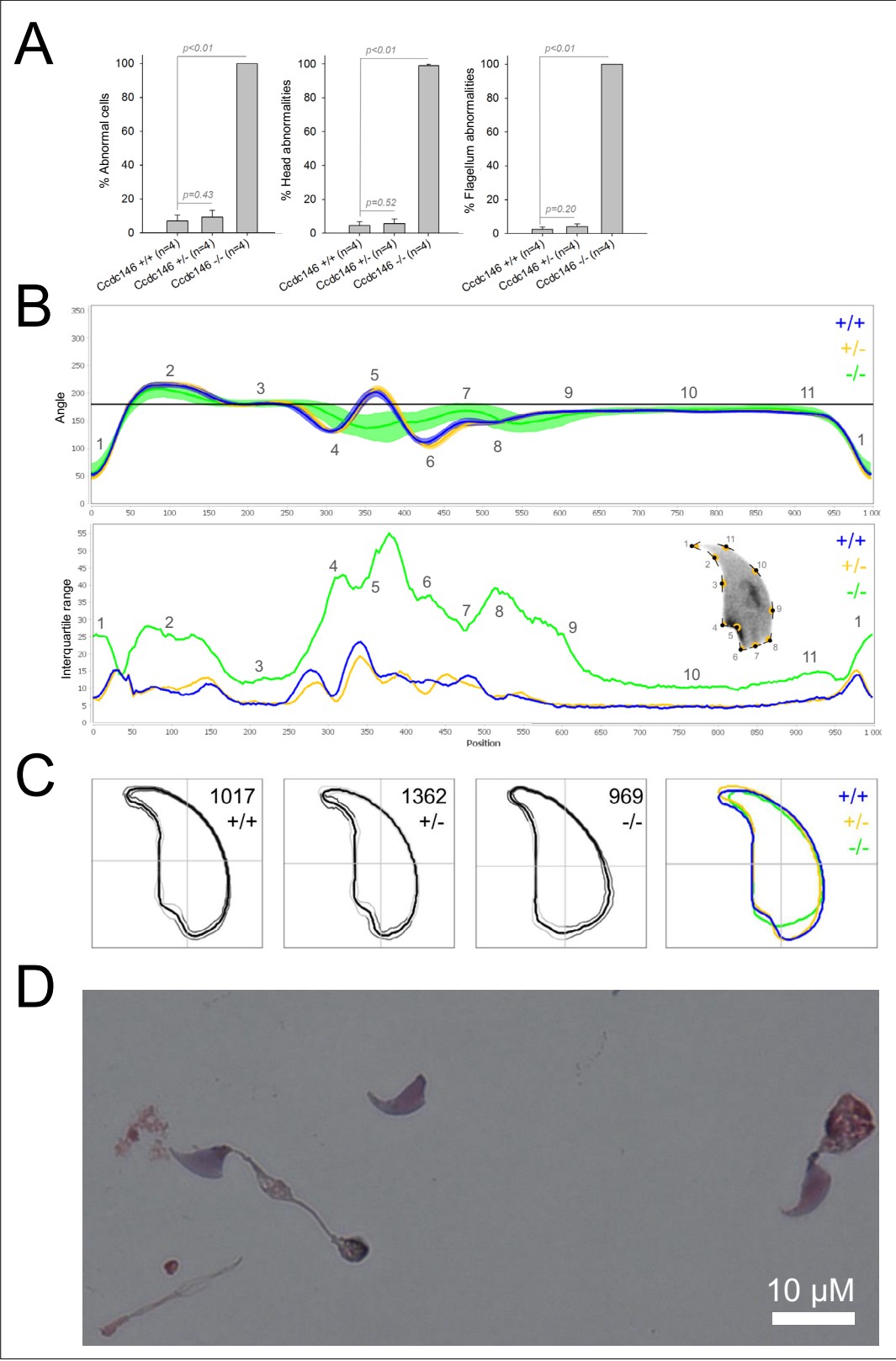

**Figure 4.** Sperm morphology analysis for the *Ccdc146* mouse strain.
 (**A**) Histogram showing proportions of morphological anomalies (mean ± SD) for each *Ccdc146* genotype. Statistical significance was assessed by applying an unpaired Welch *t*-test; *p*-values are indicated. (**B**) Angle profiles (top) and variability profiles (bottom) for *Ccdc146*<sup>+/+</sup> (blue), *Ccdc146*<sup>+/-</sup> (yellow) and *Ccdc146*<sup>-/-</sup> (green)

*Figure 4 continued on next page*

*Figure 4 continued*

mice. The x axis represents an index of the percentage of the total perimeter as measured counterclockwise from the apex of the sperm hook. The y axis corresponding to the angle profile is the interior angle measured across a sliding window centered on each index location. The y axis corresponding to the variability profile represents the Interquartile Range (IQR) (the difference between the 75th and 25th percentiles). Specific regions of the nuclei are mapped on the profile and the graphical representation (from *Skinner et al., 2019*) with: 1-tip; 2-under-hook concavity; 3-vertical; 4-ventral angle; 5-tail socket; 6-caudal bulge; 7-caudal base; 8-dorsal angle; 9–11-acrosomal curve. (**C**) Consensus nuclear outlines for each genotype alongside a merged consensus nucleus (blue = *Ccdc146⁺/⁺*, yellow = *Ccdc146⁺/⁻*, green = *Ccdc146⁻/⁻*). The numbers assigned to each consensus outline correspond to the number of nuclei processed per condition. (**D**) Optical microscopy analysis showing a typical MMAF phenotype for *Ccdc146* KO mice (scale bar, 10 µm).

one immediately visualizes flagellar defects. However, the phenotype is more complex and head defects have been associated with the flagellar phenotype since the first publications (*Coutton et al., 2018*; *Dong et al., 2018*). This is particularly the case for CFAP43/44 described in this report showing that the complete lack of these genes in KO males also strongly alters head patterning (*Coutton et al., 2018*). These head defects can actually be explained by the molecular function of the genes studied: CFAP43 is involved in intra-manchette transport (*Yu et al., 2021*) and it has been previously shown through the study of several genes such as Azh (*Mendoza-Lujambio et al., 2002*), Clip170 (*Akhmanova et al., 2005*), Rim-bp3 (*Zhou et al., 2009*), or Azi1 (*Hall et al., 2013*), that manchette modifications impact the head shape of the sperm. The implication of ARMC2 in intraflagellar transport

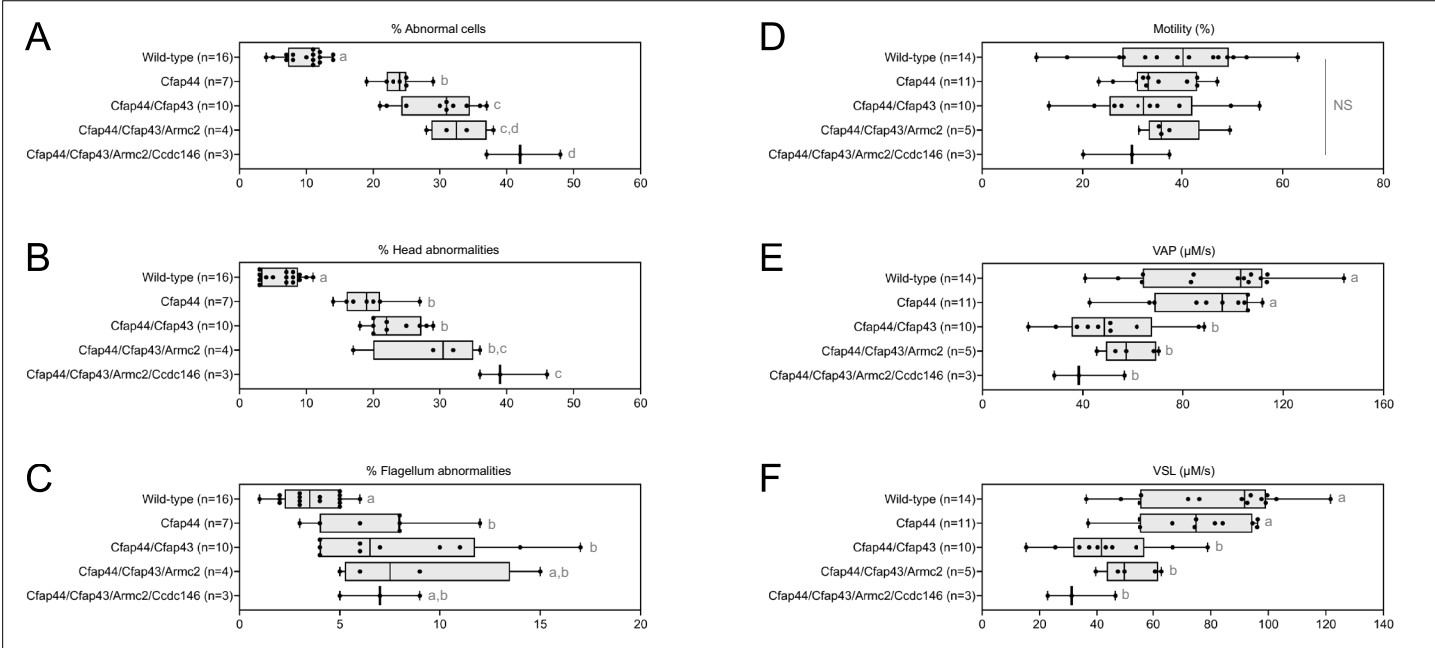

**Figure 5.** Increasing the number of heterozygous mutated genes involved in the MMAF syndrome has a drastic effect on sperm morphology and motility. (**A**) Sperm morphological defects, (**B**) head defects, (**C**) flagellum defects, showing only slight impact. Increasing the number of heterozygous gene mutations had little effect on (**D**) Percentage of motile sperm, but considerably reduced sperm motility parameters including (**E**) average path velocity (VAP), and (**F**) curvilinear velocity (VCL). All data are presented simultaneously as box-plot and individual datapoints. Statistical significance was assessed using an unpaired Welch *t*-test. Each group was compared individually with all other groups one by one. For each histogram, plots sharing different small letters represent statistically significantly differences between the groups (p < 0.05), and plots with a common letter do not present statistically significantly differences between the groups (p > 0.05). The corresponding statistical data can be found in *Figure 5—source data 2–3* and raw data can be found in *Figure 5—source data 1*.

The online version of this article includes the following source data for figure 5:

**Source data 1.** Raw data of *Figure 5*.

**Source data 2.** Statistical data linked to the Welch t-tests performed in *Figure 5D–F*.

**Source data 3.** Statistical data linked to the Welch t-tests performed in *Figure 5A–C*.

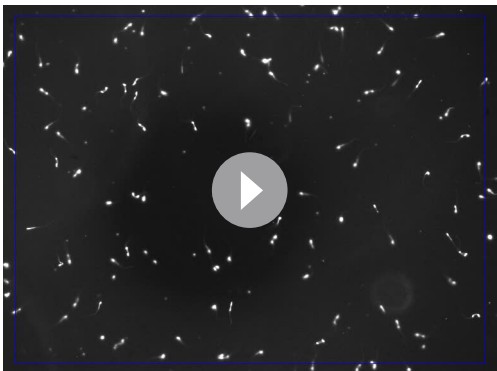

**Video 1.** Representative video of living sperm cells from wild-type mouse provided by Computer Assisted Sperm Analysis device. Sperm were introduced into a Leja slide (100 µm thick) and maintained at 37 °C during recording.

https://elifesciences.org/articles/75373/figures#video1

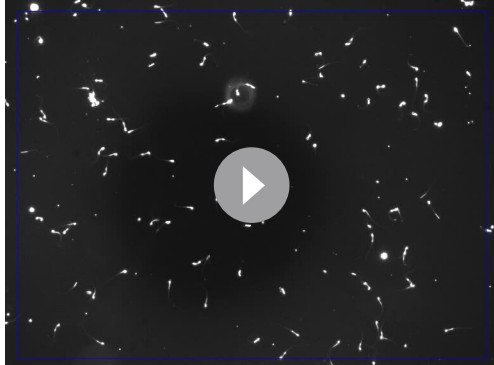

**Video 3.** Representative video of living sperm cells from mouse bearing two heterozygous mutation provided by Computer Assisted Sperm Analysis device. Sperm were introduced into a Leja slide (100 µm thick) and maintained at 37 °C during recording.

https://elifesciences.org/articles/75373/figures#video3

has also recently been shown (*Lechtreck et al., 2022*) and several previous reports associated sperm head malformations with IFT genes, like IFT20 (*Zhang et al., 2016*), IFT25 (*Liu et al., 2017*), IFT27 (*Zhang et al., 2017*), or IFT88 (*San Agustin et al., 2015*). Finally, CCDC146 was described as a centriolar proteins (*Firat-Karalar et al., 2014*) and other centriolar proteins are known to induce sperm head defects (*Hwang et al., 2021*). If the significant increase in head defects can be easily explained, we do not know why flagellar defects increase simultaneously very little.

As descriptions of animal reproductive phenotypes are becoming increasingly sophisticated and standardized (*Houston et al., 2021*), emerging tools such as the software used in this study to analyze nuclear morphology should be used to extensively study reproductive phenotypes, including studies of MMAF syndromes, to accurately document head defects.

For the first time, we were able to objectively document defects linked to MMAF using NMAS. This original method allows objective comparison of the impact of mutations on sperm head morphology. We found for *Cfap43*[-/-] and *Cfap44*[-/-] mice that sperm heads predominantly displayed a 'pepper' shape – characterized by a broadening of the base (median widths of body were 3.44 ± 0.09 µm and 3.93 ± 0.09 µm for *Cfap43*[-/-] and *Cfap44*[-/-] respectively, versus 2.95 ± 0.06 µm and 3.09 ± 0.12 µm for *Armc2*[-/-] and *ccdc146*[-/-] mice, respectively) and that the flagellum insertion notch had a reduced

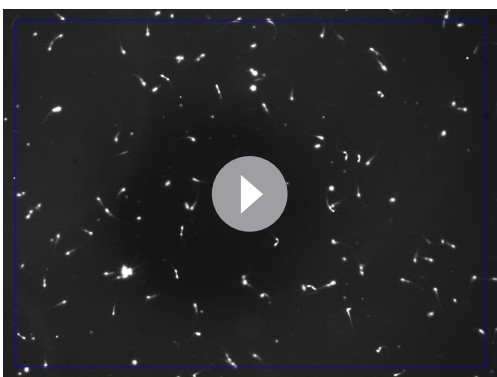

**Video 2.** Representative video of living sperm cells from mouse bearing one heterozygous mutation provided by Computer Assisted Sperm Analysis device. Sperm were introduced into a Leja slide (100 µm thick) and maintained at 37 °C during recording.

https://elifesciences.org/articles/75373/figures#video2

**Video 4.** Representative video of living sperm cells from mouse bearing three heterozygous mutation provided by Computer Assisted Sperm Analysis device. Sperm were introduced into a Leja slide (100 µm thick) and maintained at 37°C during recording.

https://elifesciences.org/articles/75373/figures#video4

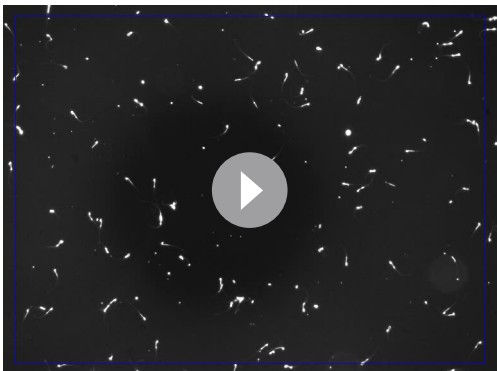

**Video 5.** Representative video of living sperm cells from mouse bearing four heterozygous mutation provided by Computer Assisted Sperm Analysis device. Sperm were introduced into a Leja slide (100 µm thick) and maintained at 37 °C during recording. https://elifesciences.org/articles/75373/figures#video5

size. *Armc2⁻ᐟ⁻* sperm show moderate enlargement extending along the entire length of the head, including the hook (length of hook increase from 7.55 ± 0.03 µm to 8.23 ± 0.05 µm between *Armc2⁺ᐟ⁺* and *Armc2⁻ᐟ⁻* mice), and an increase in circularity (increase from 0.55 to 0.61 between *Armc2⁺ᐟ⁺* and *Armc2⁻ᐟ⁻* mice) resulting in a more rounded appearance. *Ccdc146⁻ᐟ⁻* sperm presented a total disorganization of the base with complete erasure of the flagellum insertion notch, and an overall decrease in size (mean area decrease from 111.86 ± 0.41 µm² to 96.78 ± 1.08 µm² between *ccdc146⁺ᐟ⁺* and *ccdc146⁻ᐟ⁻* mice), resulting in heads with a triangular aspect characteristic of the 'claw' shape. It is worth to note that the effect on sperm head shape was gene-dependent, with a milder effect observed for *Armc2* and a stronger effect observed for *Cfap43*, *Cfap44*, and *Ccd146*. This result was notable as the flagellum phenotype of *Armc2* is as severe as that observed with mutation of the other genes. Consequently, sperm head defects are not due to failed flagellum biogenesis. Thus, compared to the other proteins, Armc2 may be less involved in functions other than flagellum biogenesis. It should be noted that NMAS analysis is complementary to the human visual analysis made from optical microscopy and does not replace it because the nature of the detected defects is slightly different. Indeed, the software can detect fine nuclear shape anomalies that are undetectable to the human eye and thus reveals morphological variations that are not retained visually by the experimenter. In this study, NMAS revealed a head morphology variation of *Armc2⁺ᐟ⁻* versus *Armc2⁺ᐟ⁺* sperm (**Figure 3B**) that had not been detected by visual analysis (**Figure 3A**). On the other hand, NMAS does not identify some abnormalities evidenced by light microscopy stains such as acrosome or flagellum insertion defects. For instance, this was illustrated by the similarity of the nuclear profiles observed between *Cfap43⁺ᐟ⁻*, *Cfap44⁺ᐟ⁻* and their wild-type counterparts (+/-), while abnormalities were clearly detected by light microscopy (**Figures 1A and 2A**). To summarize, we believe that NMAS will bring as much to the study of morphological defects of the head as CASA has brought to the study of defects of sperm motility.

Among the proteins encoded by previously identified MMAF genes, some belong to complexes involved in intraflagellar transport (*Liu et al., 2019a*; *Liu et al., 2019b*; *Chung et al., 2014*), protein degradation (*Shen et al., 2019*) or unknown processes (*Coutton et al., 2019*; *Lorès et al., 2018*) that could affect head formation, and subsequently sperm DNA. As there is a strong relationship between cytoskeletal and chromosomal effects, the question of the potential impact of the accumulation of mutations on sperm chromatin organization arises. Another interesting question would be to investigate whether the damaged heads correspond to those carrying the mutated alleles. Future studies will be eagerly awaited to elucidate the molecular basis of oligogenicity in male infertility. Therefore, we recommend that head morphology should not be overlooked when studying MMAF syndromes, despite an obvious focus on flagellum anomalies.Despite a strong focus on sperm head defects, we also showed that sperm motility was altered in the presence of ≥2 heterozygote mutations. Thus, the function of the flagellum is affected, and an absence of morphological defects should therefore not be considered synonymous with absence of functional defects. More importantly, these results support the hypothesis that idiopathic human asthenozoospermia may be due to an accumulation of heterozygous mutations in genes known to be involved in flagellum biogenesis.

To date, the inheritance pattern of isolated male infertility was only known to be Mendelian – that is, based on a single locus – and the possibility of oligogenic inheritance had not been explored. In contrast, an oligogenic etiology for female infertility has been proposed to be associated with primary ovarian insufficiency (POI) by several groups, in particular due to the identification of heterozygous mutations in several genes associated with POI (*Patiño et al., 2017*). Oligogenic inheritance has also previously been suggested in another reproductive disorder: congenital hypogonadotropic

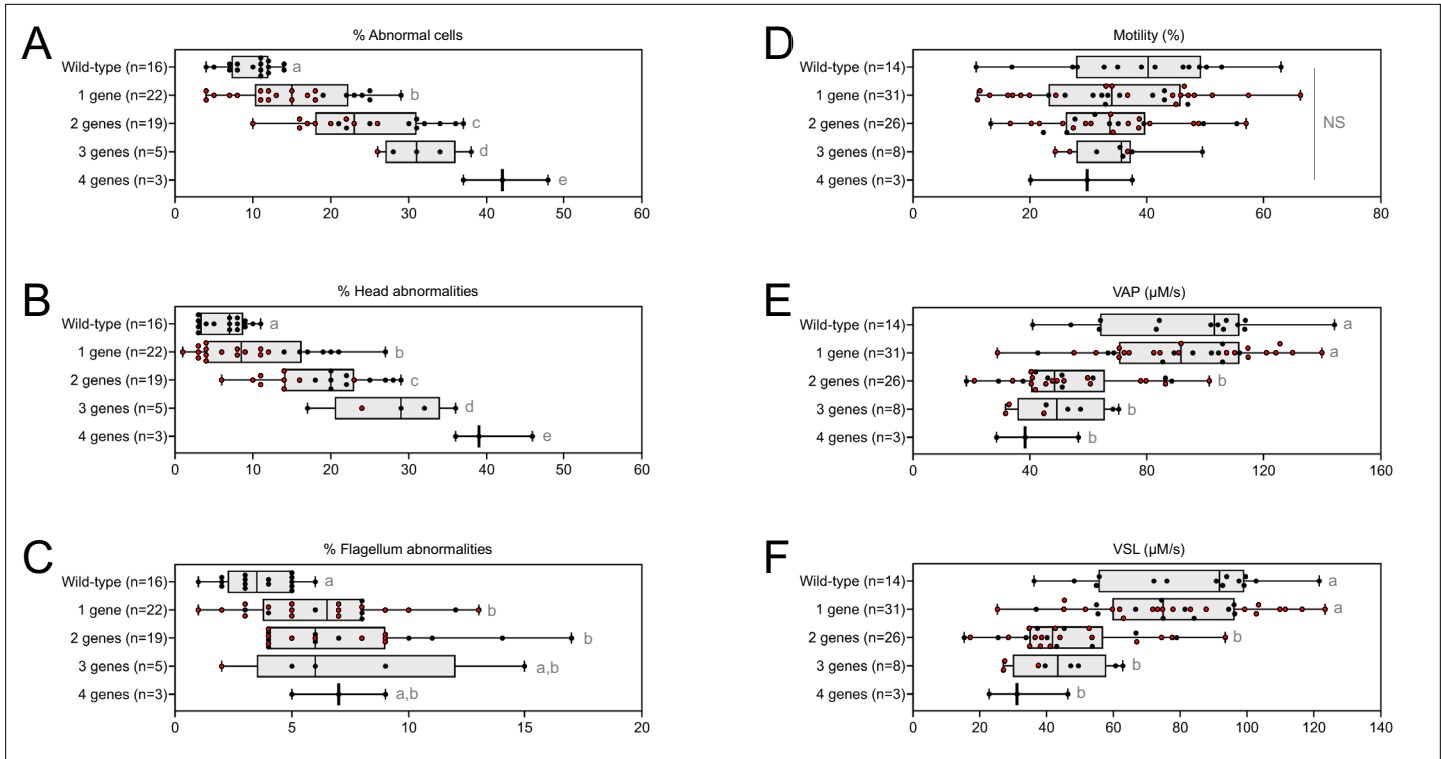

**Figure 6.** Very similar head morphology defects and decreased motility parameters, whatever the combination of mutated genes. Black dots correspond to the different combinations (2 = *Cfap43* and *Cfap44*; 3 = *Cfap43*, *Cfap44* and *Armc2*; 4 = *Cfap43*, *Cfap44*, *Armc2* and *Ccdc146*) of mutations presented in *Figure 5*; red dots correspond to alternative combinations obtained with the same four genes. The range of (**A**) sperm morphological defects, (**B**) head anomalies, and (**C**) flagellum defects was similar for black and red dots. Likewise, (**D**) percentage of motile sperm, (**E**) average path velocity (VAP) and (**F**) curvilinear velocity (VCL), showed similar ranges for black and red dots. All data are presented simultaneously as box-plot and individual datapoints. Statistical significance was assessed using unpaired Welch t-test. Each group was compared individually with all other groups one by one. For each histogram, plots sharing different small letters represent statistically significantly differences between the groups (p < 0.05), and plots with a common letter do not present statistically significantly differences between the groups (p > 0.05). Statistical significance of differences between means calculated for all black and red dots were also assessed by applying an unpaired Welch t-test and the corresponding statistical data can be found in *Figure 6—source data 2–3* and raw data can be found in *Figure 6—source data 1*.

The online version of this article includes the following source data for figure 6:

**Source data 1.** Raw data and genotypes of *Figure 6*.

**Source data 2.** Statistical data linked to the Welch t-tests performed in *Figure 6D–F*.

**Source data 3.** Statistical data linked to the Welch t-tests performed in *Figure 6A–C*.

hypogonadism (*Pitteloud et al., 2007*), as well as in several non-reproductive disorders (*Li et al., 2017*). Our results demonstrate that oligogenic inheritance may be linked to both male and female human infertility, and should therefore also be accurately measured when investigating male infertility.

In conclusion, in this article, we report the first evidence of oligogenic inheritance in altered spermatogenesis, leading to teratoasthenozoospermia. This mode of inheritance is crucial as oligogenic events could be behind the difficulties encountered by a significant proportion of infertile couples, for whom the current diagnosis is the somewhat unsatisfactory 'unexplained' or 'idiopathic' infertility. Our study was conducted in a context of teratozoospermia, a qualitative disorder of spermatogenesis. It paves the way for further studies on other male infertility disorders, including quantitative disorders, such as oligozoospermia or azoospermia for which the diagnostic yield remains very low. Current genetic tests to explore male infertility focus primarily on identifying low-frequency fully-penetrant monogenic defects, which are usually autosomal recessive and linked to the most severe cases of male infertility. However, investigation of oligogenic inheritance in the huge cohorts available should provide an estimate of the frequency of such events. These investigations could potentially identify new candidate genes involved in male infertility.

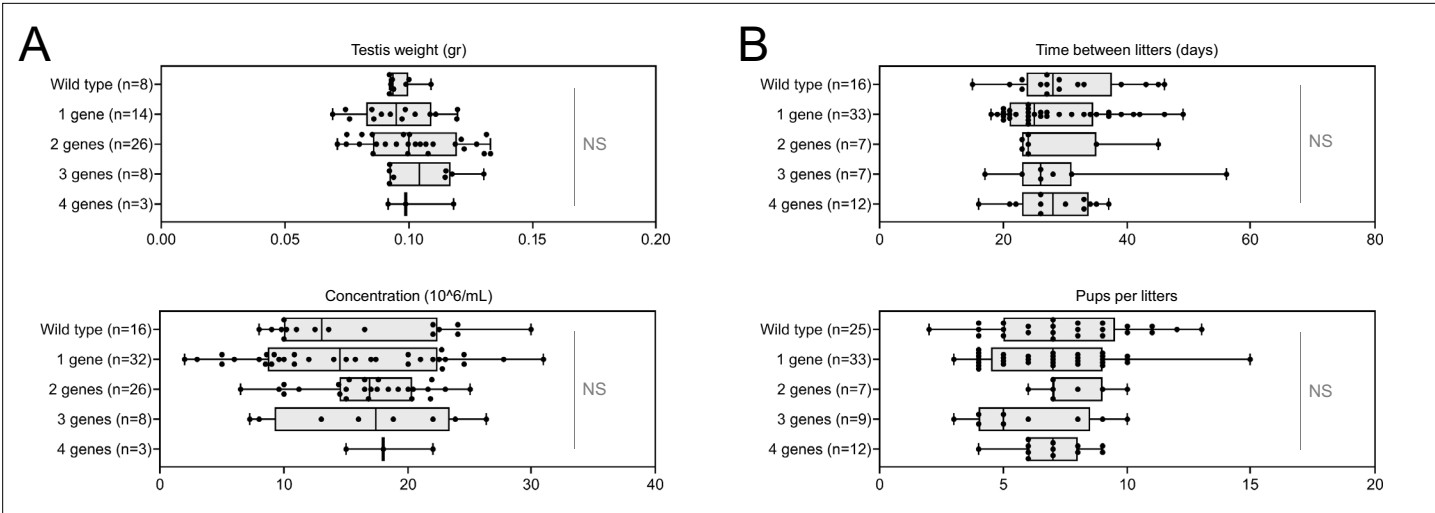

**Figure 7.** Increasing the number of mutated genes has little effect on overall fertility. (**A**) Sperm production data (**B**)Overall fertility of animals (measured as the interval between two litters and the number of pups per litter). All data are presented simultaneously as box-plot and individual datapoints. Statistical significance was assessed using unpaired Welch *t*-test. Each group was compared individually with all other groups one by one. For each histogram, plots sharing different small letters represent statistically significantly differences between the groups (p < 0.05), and plots with a common letter do not present statistically significantly differences between the groups (p > 0.05). The corresponding statistical data can be found in *Figure 7—source data 2* and raw data can be found in *Figure 7—source data 1*.

The online version of this article includes the following source data for figure 7:

**Source data 1.** Raw data and genotypes of *Figure 7*.

**Source data 2.** Statistical data linked to the Welch t-tests performed in *Figure 7B*.

**Source data 3.** Statistical data linked to the Welch t-tests performed in *Figure 7A*.

The discovery presented here is of major medical interest, and has implications for both clinical genetics and infertility management. First, the continuous and exponential characterization of new genes involved in infertility over the last decade offers the hope that, in the near future, an almost exhaustive list of genes and mutations involved in human infertility will be available. The identification of and screening for all known mutant alleles linked to male infertility at the heterozygous level should improve the diagnostic yield. Second, the discovery of multiple mutated genes will allow clinicians to provide more accurate genetic counselling to patients, and better guide them in their infertility journey. To improve patient management, future studies should look at potential correlations between patients' mutational burden and their intracytoplasmic sperm injection (ICSI) success, pregnancies achieved, and live birth rates. It will also be essential to assess the impact of mutational load on parameters known to influence these outcomes.

## Materials and methods
### Animals
Generation of *Cfap43* and *Cfap44* KO mice is described in *Coutton et al., 2018*, generation of *Armc2* KO mice is described in *Coutton et al., 2019*. CRISPR/Cas9 gene editing was used to produce *Ccdc146* KO mice (ENSMUST00000115245). To maximize the chances of producing deleterious mutations, two gRNAs located in two distinct coding exons positioned at the beginning of the targeted gene were used. For each gene, the two gRNAs (5'-CCT ACA GTT AAC ATT CGG G-3' and 5'-GGG AGT ACA ATA TTC AGT AC-3') targeting exons 2 and 4, respectively, were inserted into two distinct plasmids also containing the Cas9 sequence. The Cas9 gene was driven by a CMV promoter and the gRNA and its RNA scaffold by a U6 promoter. Full plasmids (pSpCas9 BB-2A-GFP (PX458)) containing the specific sgRNA were ordered from Genscript (https://www.genscript.com/crispr-cas9-protein-crRNA.html). Both plasmids were co-injected into the zygotes' pronuclei at a concentration of 2.5 ng/ml. It should be noted that the plasmids were injected as delivered by the supplier, thus avoiding the need to perform in vitro production and purification of Cas9 proteins and sgRNA. Several mutated

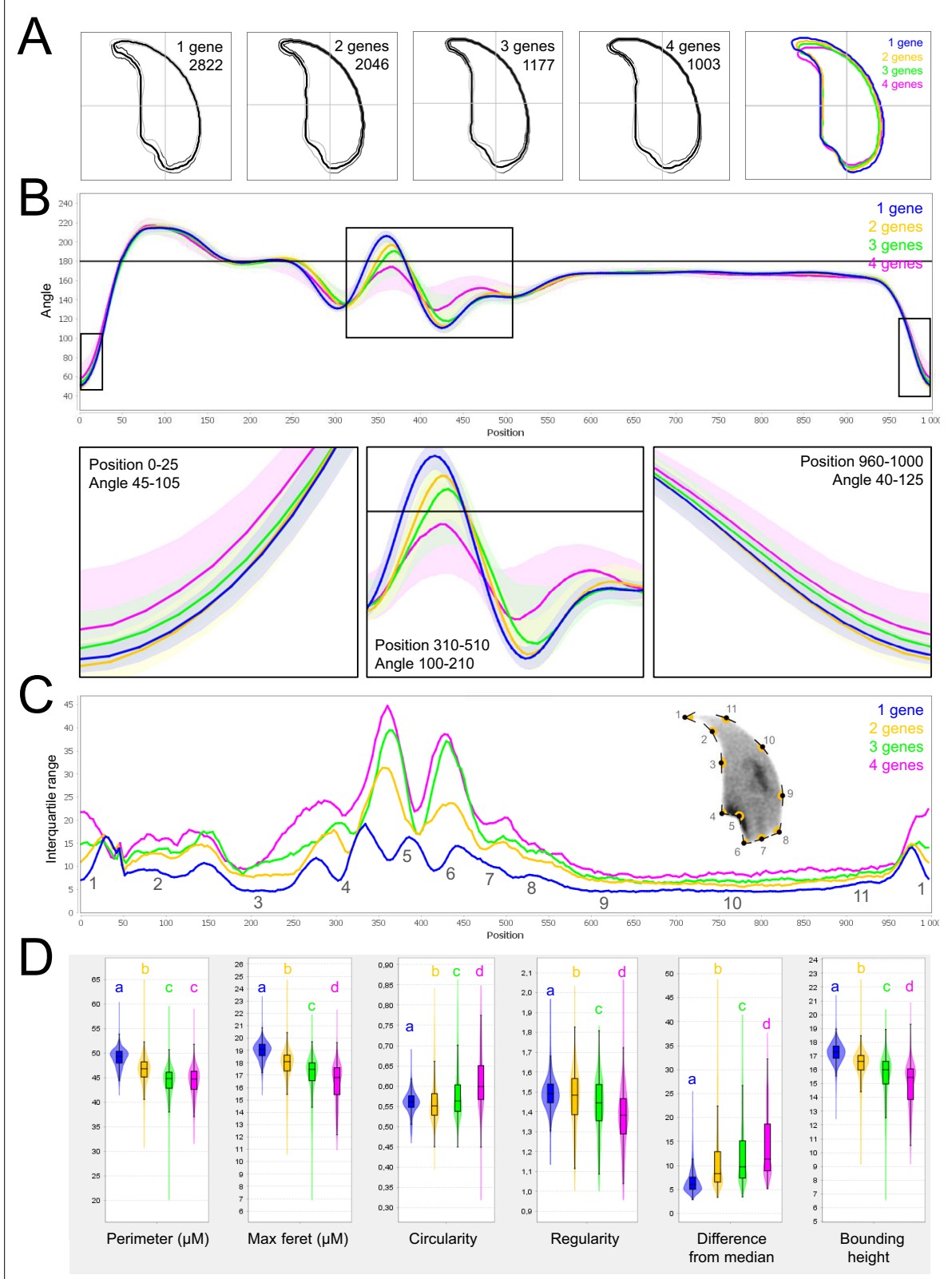

**Figure 8.** Fine nuclear morphology analysis of multi-mutant mice with one to four heterozygous mutations.
 (**A**) Consensus nuclear outlines for each strain alongside a merged consensus nucleus (blue = one mutation, yellow = two mutations, green = three mutations and pink = four mutations). The numbers assigned to each consensus outline correspond to the number of nuclei processed per condition.
(**B**) Angle profiles for each strain focusing (black boxes) on positions of specific interest. The x axis represents an index of the percentage of the total

*Figure 8 continued on next page*

*Figure 8 continued*

perimeter, as measured counterclockwise from the apex of the sperm hook. The y axis shows the interior angle measured across a sliding window centered on each index location. (**C**) Variability profiles for each strain. The x axis is the same as for the angle profile, and the y axis represents the Interquartile Range (IQR) (the difference between the 75th and 25th percentiles). Specific regions of the nuclei are mapped on the profile and the graphical representation (from *Skinner et al., 2019*), with: 1-tip; 2-under-hook concavity; 3-vertical; 4-ventral angle; 5-tail socket; 6-caudal bulge; 7-caudal base; 8-dorsal angle; 9–11-acrosomal curve. (**D**) Violin plots of nuclear parameters for each strain. Statistical significance of differences between populations were assessed by the software, applying a Mann-Whitney U test; significantly different populations are identified by distinct letters.

animals were obtained with different insertions/deletions spanning a few nucleotides. All the mutations obtained induced a translational frameshift expected to lead to complete absence of the protein or production of a truncated protein. The reproductive phenotype of two mutated lines was analyzed; homozygous males were infertile and displayed the same MMAF phenotype. For this study, a strain with a 4 bp deletion in exon 2 was used (c.164_167delTTCG).

For each KO strain, mice were maintained in the heterozygous state, and males and females were crossed to produce animals for subsequent generations. Heterozygous and homozygous animals were selected following PCR screening, the primers used for each strain are indicated in *Appendix 1— figure 1B*. To produce multi-heterozygous animals, homozygous females were crossed with heterozygous males.

All animal procedures were conducted according to a protocol approved by the local Ethics Committee (ComEth Grenoble No. 318), by the French government (ministry agreement number #7,128 UHTA-U1209-CA), and by the Direction Générale de la Santé (DGS) for the State of Geneva. Guide RNA, TracrRNA, ssDNA, and Cas9 were purchased from Integrated DNA Technologies. Pronuclear injection and embryo transfer were performed by the Transgenic Core facility at the Faculty of Medicine, University of Geneva. All genotypes were obtained by conventional interbreeding.

## Reproductive phenotyping

All adult male mice used were between 10 and 12 weeks old. After sacrifice by cervical dislocation, the testes were isolated and weighed, and sperm from caudae epididymides were allowed to swim for 10 min at 37 °C in 1 ml of M2 medium (Sigma-Aldrich, L'Isle d'Abeau, France). The sperm concentration was measured and adjusted before CASA analysis. An aliquot of sperm suspension was immediately placed in a 100 µm deep analysis chamber (Leja Products B.V., Nieuw-Vennep, the Netherlands), and sperm motility parameters were measured at 37 °C using a sperm analyzer (Hamilton Thorn Research, Beverley, MA, USA). The settings used for analysis were: acquisition rate: 60 Hz; number of frames: 45; minimum contrast: 50; minimum cell size: 5; low static-size gate: 0.3; high static-size gate: 1.95; low static-intensity gate: 0.5; high static-intensity gate: 1.3; minimum elongation gate: 0; maximum elongation gate: 87; magnification factor: 0.7. Motile sperm were defined by an average path velocity (VAP) >1, and progressive sperm motility was defined by VAP >30 and average path straightness >70. A minimum of 200 motile spermatozoa were analyzed in each assay. Remaining sperm was rinsed with PBS-1X, centrifuged for 5 min at 500 g, spread on slides and allowed to dry at room temperature. Samples were then fixed in Ether/Ethanol 1:1 for Harris-Schorr staining (to assess overall morphology) or in 4% paraformaldehyde for DAPI staining (to assess nuclear morphology).

Morphology was visually assessed on a Nikon Eclipse 80i microscope equipped with a Nikon DS-Ri1 camera with NIS-ElementsD (version 3.1.) software by trained experimenters. At least 200 spermatozoa were counted per slide at a magnification of ×1000. Cells are classified as abnormal when they bear at least one morphological defect, either on the head or the flagellum. Normal head morphology is defined by a typical murine overall shape with a pointy kook tip, a well-defined flagellum insertion notch in continuation of a smooth central region, a prominent caudal buldge and a dorsal region without notches. Normal flagellum must be continuous, of regular size and caliber, without angulation or excessive curling. Examples of normal and abnormal morphologies are provided in *Appendix 1— figure 7*.

## Analysis of nuclear morphology

Nuclear morphology was precisely evaluated by NMAS (version 1.19.2, https://bitbucket.org/bmskinner/nuclear_morphology/wiki/Home), according to the analysis method described in *Skinner et al., 2019*. The software processed images of DAPI-stained nuclei captured with a Zeiss Imager Z2

microscope, using a CoolCube 1 CCD camera, with a 100 x/1.4 Zeiss objective and Neon software (MetaSystems, Altlussheim, Germany). Nucleus detection settings were Kuwahara kernel: 3, and flattening threshold: 100, for preprocessing; canny low threshold: 0.5, canny high threshold: 1.5, canny kernel radius: 3, canny kernel width: 16, gap closing radius: 5, to find objects; and min area: 1 000, max area: 10 000, min circ: 0.1, max circ: 0.9, for filtering. After acquisition of images of nuclei, landmarks were automatically identified using a modification of the Zahn-Roskies transform to generate an angle profile from the internal angles measured around the periphery of the nuclei. Angles were measured at every mouse pixel around the original shape. This method combines data from every possible polygonal approximation into a single unified trace, from which landmark features can be detected (under-hook concavity, tail socket, caudal bulge and base, acrosomal curve, etc.). Angle profiles are presented as angle degrees according to the relative position of each pixel along the perimeter, and variability profiles use the interquartile range (IQR, difference between the third and first quartile) as a dispersion indicator to measure the variability of values obtained for each point. Sperm shape populations were then clustered using a hierarchical ward-distance method without reduction, based on angle profiles.

## Statistical analysis

The statistics relating to nuclear morphology presented in *Figure 8* and *Appendix 1—figure 4* were automatically calculated by the Nuclear Morphology Analysis Software. This analysis relied on a Mann-Whitney U test with Bonferroni multiple testing correction. p-values were considered significant when inferior to 0.05.

All other data were treated with R software (version 3.5.2). Histograms show mean ± standard deviation, and statistical significance of differences was assessed by applying an unpaired Welch *t*-test. Statistical tests with two-tailed p-values ≤ 0.05 were considered significant.

## Acknowledgements

We sincerely thank Roxane DOMINGUEZ and Aurélien SIMON for their assistance with the software and IT solutions. We are also grateful to Charlotte GUYOT and Marlène GANDULA for technical assistance.

## Additional information

### Funding

| Funder | Grant reference number | Author |
| --- | --- | --- |
| Agence Nationale de la Recherche | ANR-19-CE17-0014 | Pierre F Ray Christophe Arnoult |
| Agence Nationale de la Recherche | ANR-21-CE17-0007 | Guillaume Martinez Charles Coutton |

The funders had no role in study design, data collection and interpretation, or the decision to submit the work for publication.

### Author contributions

Guillaume Martinez, Investigation, Conceptualization, Funding acquisition, Methodology, Supervision, Writing - original draft, Writing - review and editing; Charles Coutton, Investigation, Conceptualization, Formal analysis, Funding acquisition, Methodology, Resources, Supervision, Writing - original draft, Writing - review and editing; Corinne Loeuillet, Caroline Cazin, Jana Muroňová, Magalie Boguenet, Emeline Lambert, Magali Dhellemmes, Geneviève Chevalier, Jean-Pascal Hograindleur, Charline Vilpreux, Yasmine Neirijnck, Zine-Eddine Kherraf, Jessica Escoffier, Funding acquisition; Serge Nef, Funding acquisition, Resources; Pierre F Ray, Investigation, Formal analysis, Methodology, Resources, Writing - original draft; Christophe Arnoult, Investigation, Conceptualization, Formal analysis, Methodology, Project administration, Resources, Supervision, Writing - original draft, Writing - review and editing

## Author ORCIDs

Guillaume Martinez http://orcid.org/0000-0002-7572-9096
Jessica Escoffier http://orcid.org/0000-0001-8166-5845
Christophe Arnoult http://orcid.org/0000-0002-3753-5901

## Ethics

All animal procedures were conducted according to a protocol approved by the local Ethics Committee (ComEth Grenoble No. 318), by the French government (ministry agreement number #7128 UHTA-U1209-CA), and by the Direction Générale de la Santé (DGS) for the State of Geneva.

## Decision letter and Author response

Decision letter https://doi.org/10.7554/eLife.75373.sa1
Author response https://doi.org/10.7554/eLife.75373.sa2

---

# Additional files

## Supplementary files

• Transparent reporting form

## Data availability

Figure 5 - Source Data 1, Figure 6 - Source Data 1 and Figure 7 - Source Data 1 contain the numerical data used to generate the figures.

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

# Appendix 1

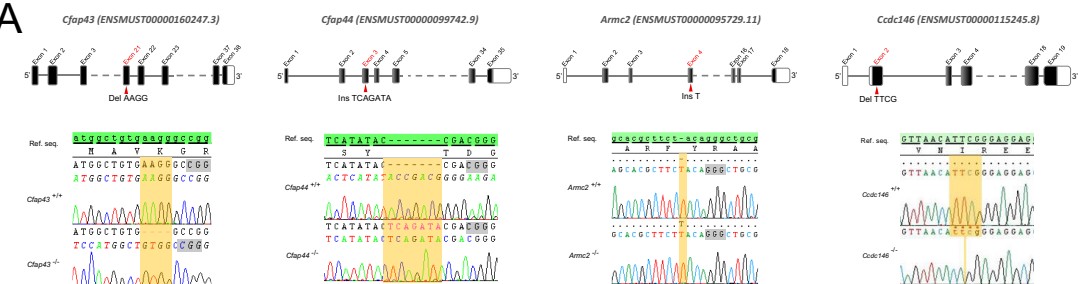

| Primer name | Sequence (5'->3') | Tm (°C) |
|---|---|---|
| Wdr52_WT-F | ACACCACAGCTGACTCATATACC | 59.87 |
| Wdr52_KO-F | CCACAGCTGACTCATATACTCAGA | 59.42 |
| Wdr52_U-R | TGACAGTTACTGAGAGAGAGCG | 59.26 |
| Wdr96_WT-F | CCTTTAGAGCCCGGCCCTT | 62.4 |
| Wdr96_KO-F | CCTTTAGAGCCCGGCCAC | 61.2 |
| Wdr96_U-R | GACCGACCATAGCGTTGTCT | 58.7 |
| Ccdc146_Exon4-F | GCAGCTGCTCAAGTACCAAA | 58.76 |
| Ccdc146_Exon4-R | CCGGGGACTCACCTAGTCA | 60.00 |
| Armc2_WT-F | GGCCCGAGCACGCTTCTA | 61.00 |
| Armc2_WT-R | TTCATGTAAGAACTATCCAGGACCA | 57.8 |
| Armc2_KO-F | TGGGACGCAGCCCTGTAA | 59.8 |
| Armc2_KO-R | AACCCAAAGCTCCAGCATCTC | 59.3 |

**Appendix 1—figure 1.** Genetic data associated with strains creation. (**A**) Location of mutations and electropherograms from Sanger sequencing of mutated forms of murine *Cfap43*, *Cfap44*, *Armc2*, and *Ccdc146* compared to their respective reference sequences. We confirmed a 4 bp deletion in *Cfap43* exon 21 (delAAGG), a 7 bp insertion in *Cfap44* exon 3 (InsTCAGATA), a 1 bp insertion (InsT) in *Armc2* exon 4, and a 4 bp deletion in *Ccdc146* exon 2 (delTTCG). Red arrows indicate the CRISPR/Cas9 targeting sequence, mutations are highlighted in yellow, and the gray boxes indicate the position of the protospacer-adjacent motif (PAM) sequences used during mutagenesis. (**B**) List of the primers used for PCR screening of each strain.

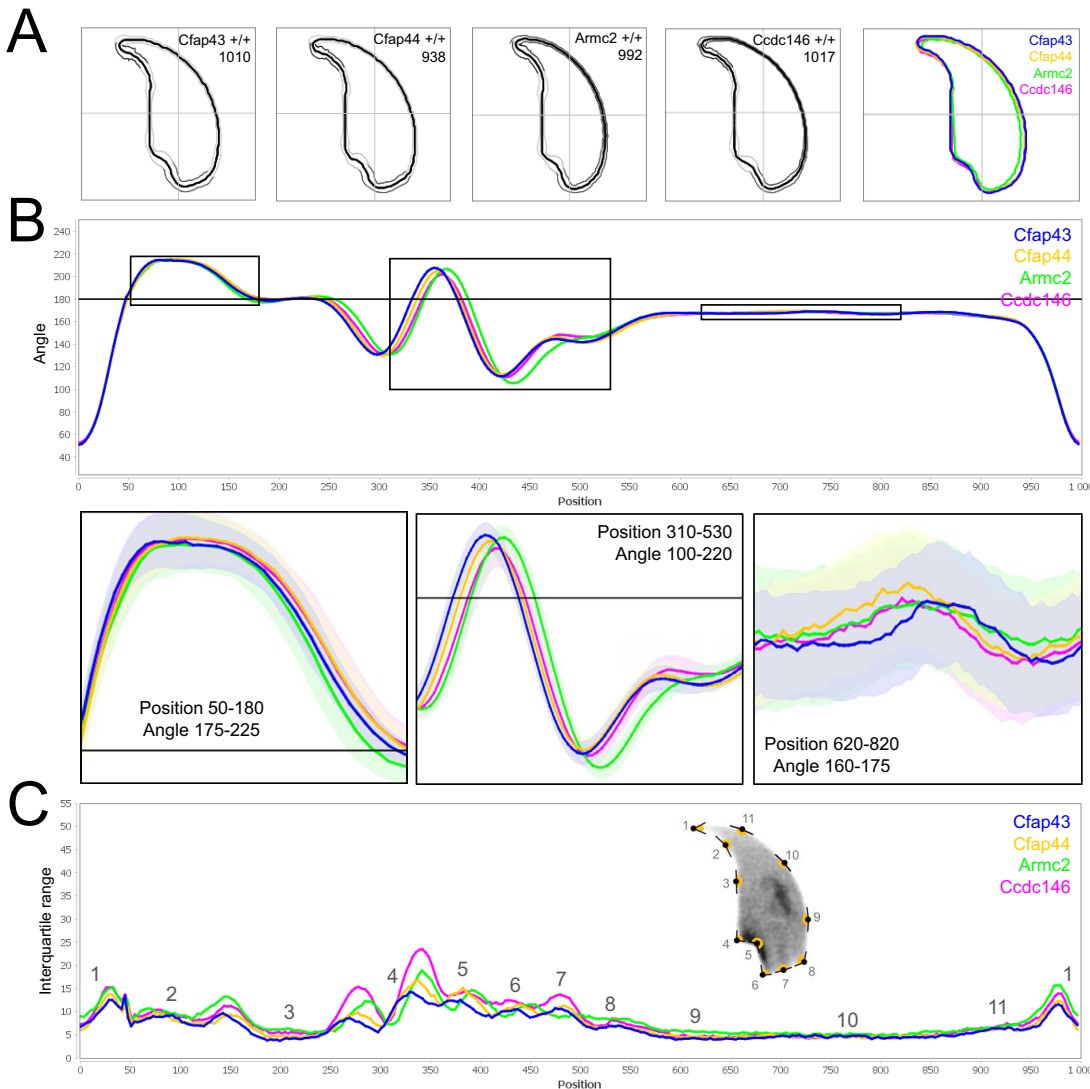

**Appendix 1—figure 2.** Comparison of fine nuclear morphology for wild-type animals produced from crosses of all strains studied. (**A**) Consensus nuclear outlines for each strain alongside a merged consensus nucleus (blue = *Cfap43*, yellow = *Cfap44*, green = *Armc2* and pink = *Ccdc146*). The numbers assigned to each consensus outline correspond to the number of nuclei processed per condition. (**B**) Angle profiles for each strain focusing (black boxes) on positions of specific interest. The x axis represents an index of the percentage of the total perimeter, as measured counterclockwise from the apex of the sperm hook. The y axis shows the interior angle measured across a sliding window centered on each index location (**C**) Variability profiles for each strain. The x axis is the same as for the angle profile, and the y axis represents the Interquartile Range (IQR) (the difference between the 75th and 25th percentiles). Specific regions of the nuclei are mapped on the profile and the graphical representation (from *Skinner et al., 2019*), with: 1-tip; 2-under-hook concavity; 3-vertical; 4-ventral angle; 5-tail socket; 6-caudal bulge; 7-caudal base; 8-dorsal angle; 9–11-acrosomal curve.

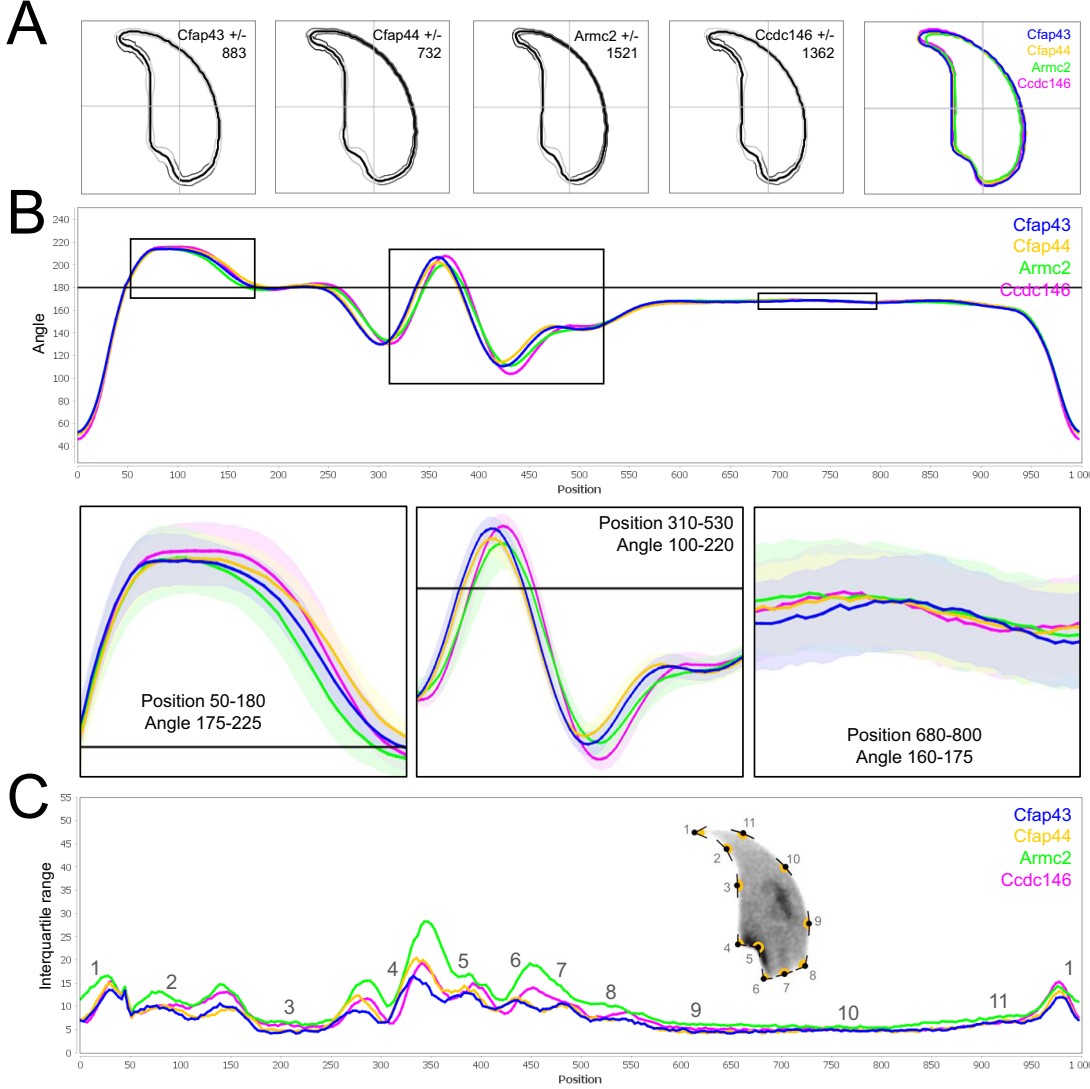

**Appendix 1—figure 3.** Comparison of fine nuclear morphology for heterozygous animals produced during crosses of all strains. (**A**) Consensus nuclear outlines for each strain alongside a merged consensus nucleus (blue = *Cfap43*, yellow = *Cfap44*, green = *Armc2*, and pink = *Ccdc146*). The numbers assigned to each consensus outline correspond to the number of nuclei processed per condition. (**B**) Angle profiles for each strain focusing (black boxes) on positions of specific interest. The x axis represents an index of the percentage of the total perimeter, as measured counterclockwise from the apex of the sperm hook. The y axis shows the interior angle measured across a sliding window centered on each index location (**C**) Variability profiles for each strain. The x axis is the same as for the angle profile, and the y axis represents the Interquartile Range (IQR) (the difference between the 75th and 25th percentiles). Specific regions of the nuclei are mapped on the profile and the graphical representation (from *Skinner et al., 2019*) with: 1-tip; 2-under-hook concavity; 3-vertical; 4-ventral angle; 5-tail socket; 6-caudal bulge; 7-caudal base; 8-dorsal angle; 9–11-acrosomal curve.

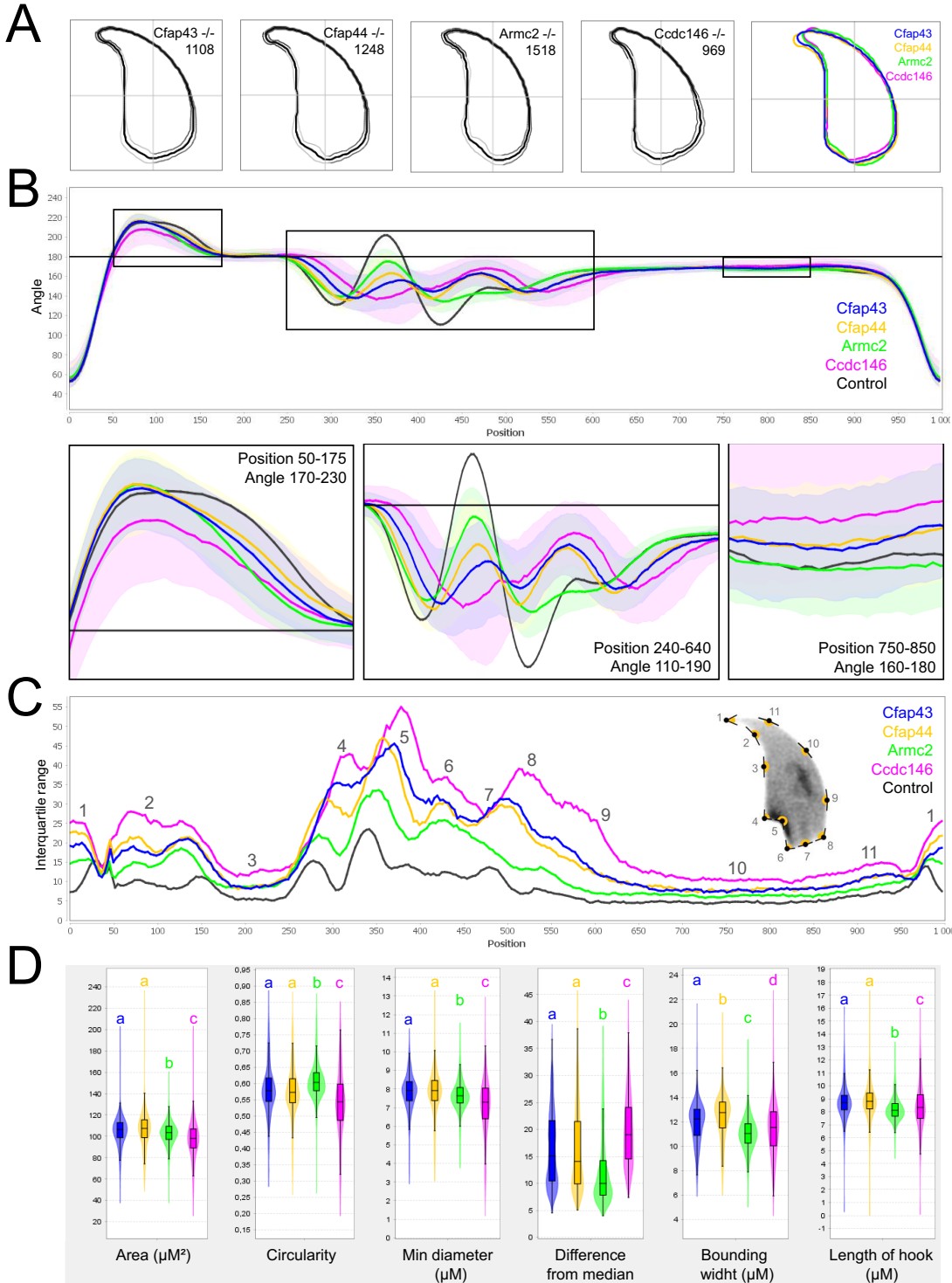

**Appendix 1—figure 4.** Comparison of fine nuclear morphology for sperm heads from KO and WT animals. (**A**) Consensus nuclear outlines for each KO strain alongside a merged consensus nucleus, and (far right) superimposition of the outlines of the four strains (blue = *Cfap43*, yellow = *Cfap44*, green = *Armc2*, and pink = *Ccdc146*). The numbers assigned to each consensus outline corresponds to the number of nuclei processed per condition. (**B**) Angle profiles for each strain alongside an angle profile for wild-type mice (black), focusing (black boxes) on positions of specific interest. The x axis represents an index of the percentage of the total perimeter, as measured counterclockwise from the apex of the sperm hook. The y axis corresponds to the interior angle measured across a sliding window centered on each index location (**C**) Variability profiles for each strain. The x axis

*Appendix 1—figure 4 continued on next page*

*Appendix 1—figure 4 continued*
is the same as for the angle profile, and the y axis represents the Interquartile Range (IQR) value (the difference between the 75th and 25th percentiles). Specific regions of the nuclei are mapped on the profile and the graphical representation (from *Skinner et al., 2019*), with: 1-tip; 2-under-hook concavity; 3-vertical; 4-ventral angle; 5-tail socket; 6-caudal bulge; 7-caudal base; 8-dorsal angle; 9–11-acrosomal curve. (**D**) Violin plots presenting mean data for representative nuclear parameters associated with each gene. Statistical significance of differences between populations was assessed by the NMAS software, applying a Mann-Whitney U test. Significantly different populations are identified by distinct letters.

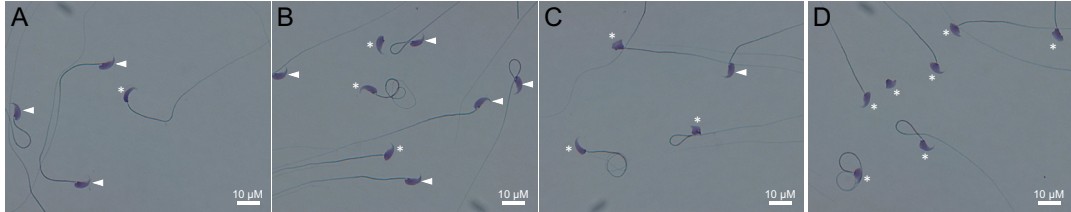

**Appendix 1—figure 5.** Light microscopy pictures of spermatozoa from mice bearing one mutation (**A**), two mutations (**B**), three mutations (**C**) or four mutations (**D**). Sperm with normal morphology are marked with a white arrow and those with a morphological abnormality are marked with a white asterisk.

| Consensus | | | | | | | | | | | % abnormal cells |
|---|---|---|---|---|---|---|---|---|---|---|---|
| Head shape | Normal | Drop | Pepper | Claw | Sickle | Hammer | Cross | Cap | Lightning | Needle | |
| No mutation | 98.33 | 0.53 | 0.00 | 0.82 | 0.07 | 0.18 | 0.00 | 0.00 | 0.00 | 0.07 | 1.67 |
| One het. Mutation | 95.55 | 1.48 | 0.11 | 1.83 | 0.30 | 0.34 | 0.11 | 0.04 | 0.04 | 0.19 | 4.45 |
| Two het. Mutations | 76.74 | 3.81 | 1.03 | 5.47 | 1.56 | 5.38 | 1.86 | 2.00 | 0.68 | 1.47 | 23.26 |
| Three het. Mutations | 66.60 | 6.54 | 3.02 | 5.53 | 1.81 | 6.74 | 2.92 | 2.31 | 1.31 | 3.22 | 33.40 |
| Four het. mutations | 44.71 | 3.35 | 6.26 | 8.53 | 3.24 | 7.34 | 10.04 | 4.64 | 3.13 | 8.75 | 55.29 |

**Appendix 1—figure 6.** Consensus and percentage distribution of the distinct sperm populations classified based on nuclear morphology and according to the number of heterozygous mutations.

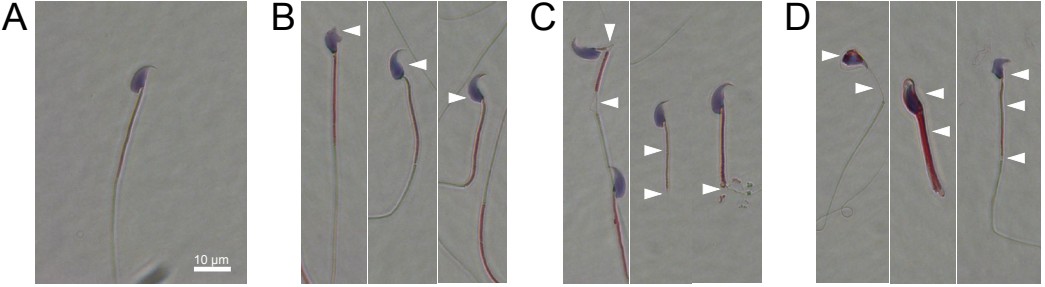

**Appendix 1—figure 7.** Light microscopy pictures of Harris-Schorr stained spermatozoa with (**A**) typical morphology, (**B**) head abnormalities, (**C**) flagellum abnormalities or (**D**) head and flagellum abnormalities. Defects are pointed by white arrows.

