## [Editor Report]

This study provides insights into the detrimental effect of accumulative heterozygous mutations on sperm abnormalities. By breeding a series of knockout strains known to cause multiple flagellar defects, the authors demonstrated that such variations at two (digenic) or more loci (oligogenic) can contribute to sperm abnormalities in the head. These findings are significant in that they implicate oligogenic inheritance as a possible cause of unexplained male infertility.

---

## [Decision Letter]

**Decision letter after peer review:**

Thank you for submitting your article "Oligogenic heterozygous inheritance of sperm abnormalities in mouse" for consideration by *eLife*. Your article has been reviewed by 2 peer reviewers, including Jean-Ju Chung as Reviewing Editor and Reviewer #1, and the evaluation has been overseen by Molly Przeworski as the Senior Editor. The following individual involved in review of your submission has agreed to reveal their identity: Donald F Conrad (Reviewer #2).

Essential revisions:

1) It is suggested that the authors use novelty claims to a minimum and interpret/describe their data quantitatively using numeric comparison and avoid using subjective interpretations.

2) If already taken, videos showing the whole images of sperm morphology and motility of the lines harboring multiple heterozygous mutations will improve the readability and interpretation of this study. Minimally, more images of representative sperm are required.

3) Please clarify human vs. computer (NMAS) interpretation and how human interpretation is implemented in text.

4) It is not clear whether the head shape difference by the oliogenic inheritance can be considered to sperm defects to cause male infertility. It is recommended that the authors to select sperm with abnormal forms from the mutant animals and demonstrate that they have certain defects for fertilization.

*Reviewer #1 (Recommendations for the authors):*

1. The new theory in this study is quite interesting and current results support the possibility. However, it is strongly suggested that the authors use novelty claims to a minimum and interpret/describe their data quantitatively using numeric comparison and avoid using subjective interpretations such as 'a breakthrough in sperm morphology assessment (line 245), strong broadening (line 249), such a strong impact on head morphology (line 254)'.

2. To improve the audience's readability and interpretation, videos showing the whole images of sperm morphology and motility of the lines harboring multiple heterozygous mutations will be required. For example, the readers will want to see the morphology and motility of representative sperm cells of1het, 2het, 3 het and 4 het.

3. Although the suggested concept is appealing, the current dataset shows that heterozygous mutations on four MMAF-causing genes, CFAP43, CFAP44, ARMC2, and CCDC146, do not result in male fertility issues. Thus, it is not clear how critical oligogenic mutations are in male fertility. In addition, the authors do not present data to explain why accumulated mutations on MMAF causing genes causes accumulative head morphology defects but not flagellar morphology. It is the area that need to be addressed with more experimental evidence.

For example, head morphology defects was shown in Danh2-KO sperm (PMID: ) in which the protein levels or localizations of other axonemal components are demonstrated to be altered. While it is a widely accepted idea that the cytoplasm (mRNAs and proteins) is shared during spermatogenesis by forming syncytium (thus the protein levels of haploid spermatids of heterozygous males are typically in the same levels as those from wild type animals), demonstrating the protein levels and/or localization of CFAP43 CFAP44, ARMC2, and CCDC146 in the sperm bearing these multiple heterozygous mutations will be critical.

4. From the comparison of +/- and WT sperm head shape in each mouse model, the authors represent that head morphology of CFAP43 and CFAP44+/- sperm are significantly abnormal than WT sperm (Figure 1-2, B). And the proportion of head abnormal sperm between ARMC2+/- males and WT males are not significantly different (Figure 3B). However, the authors' NMAS analyses show that the shapes (angle at each points) of CFAP43+/- and CFAP44+/- sperm head are very similar to those of WT sperm (Figure 1-2, C and D). Yet, ARMC2 +/- sperm show clear difference compared to WT sperm (Figure 3C and D). This discrepancy should be explained clearly.

5. Lines 241-243: Although the authors represent differences on sperm proportion with head defects, it is quite unclear whether that head shape difference can be considered to "sperm defects" to cause male infertility. In addition, authors should clearly demonstrate why sperm head shape is more affected by the accumulated "MMAF-causing" gene mutation as it will cast the question what is the 'definition' of MMAF.

The authors interprets that their observation that accumulation of heterozygous mutations leads to marked head defects without obvious morphological flagellar defects suggest 'head shape is more prone to collapse than flagellum biogenesis'. This is highly speculative – those MMAF gene products could simply participate in head morphogenesis before targeting, and/or the less efficiently transporting/targeting of the products simply could remain in the cell body, thus interfereing streamlined shaping of the head (as suggested in 260-262). Just as the comment for the point #3, demonstrating the protein levels and/or localization of CFAP43 CFAP44, ARMC2, and CCDC146 in the sperm bearing these multiple heterozygous mutations will be critical.

6. In general, it is suggested for the authors to clarify the claims/conclusion supported by their data vs. speculation throughout the text.

---

## [Author Response]

Essential revisions:1) It is suggested that the authors use novelty claims to a minimum and interpret/describe their data quantitatively using numeric comparison and avoid using subjective interpretations.

All results are now quantitatively described. Moreover, we rephrased sentences to avoid novelty claims.

2) If already taken, videos showing the whole images of sperm morphology and motility of the lines harboring multiple heterozygous mutations will improve the readability and interpretation of this study. Minimally, more images of representative sperm are required.

We now enclose videos of sperm of the different genotypes (Videos 1-5). We provide also more images of sperm defects (Appendix 1-Figure 5).

3) Please clarify human vs. computer (NMAS) interpretation and how human interpretation is implemented in text.

We added a paragraph in the Discussion section dealing with the human VS computer interpretation (lines 338-350).

4) It is not clear whether the head shape difference by the oliogenic inheritance can be considered to sperm defects to cause male infertility. It is recommended that the authors to select sperm with abnormal forms from the mutant animals and demonstrate that they have certain defects for fertilization.

Despite we were unable to show at this stage an obvious infertility phenotype, we clearly show a significant increase of proportion of abnormal sperm and decreased sperm motility. Both defective sperm motility and abnormal sperm head morphology are well recognized factors affecting male fertility and should be assessed for each patient according to WHO. Concerning sperm motility, it has been clearly shown that the percentage of conception depends on the total number of motile sperm within the semen (Publicover and Barratt, 2011^doi:10.1093/molehr/gar048^). Concerning abnormal head morphology, there are numerous reports of a significant correlation between morphology and infertility and it is accepted that any increase in any sperm abnormality should be regarded as a possible cause of decreased fertility, and that precise analyses of sperm abnormalities is a useful approach for diagnosis or research purposes (Andrade-Rocha et al., 2001^PMID:11441683^; Chemes et al., 2003^doi:10.1093/humupd/dmg034^; Menkveld et al., 2011^doi:10.1038/aja.2010.67^; Mitchell *et al.,* 2015^doi:10.1111/and.12249^; Auger et al., 2016^doi:10.1093/humrep/dev251^). For these reasons, we think that our results will interest all andrologists, even if a clear phenotype of infertility was not shown.

See also our response concerning this point below: reviewer 1 point 3. We have added a paragraph addressing this point (lines 234-246).

Reviewer #1 (Recommendations for the authors):1. The new theory in this study is quite interesting and current results support the possibility. However, it is strongly suggested that the authors use novelty claims to a minimum and interpret/describe their data quantitatively using numeric comparison and avoid using subjective interpretations such as 'a breakthrough in sperm morphology assessment (line 245), strong broadening (line 249), such a strong impact on head morphology (line 254)'.

We thank the reviewer for her suggestion. We have modified the results and Discussion sections following her recommendations and provided numbers where relevant.

Modifications involved the following lines:

Quantitative description of the results:

line 111 (increases percentages), line 126 (variation degrees), lines 130-131 (IQR variability), lines 173-177 (anomalies percentages), lines 179 (statistics), , lines 182-184 (numbers and statistics), line 186 (percentages), line 187-188 (numbers), lines 188-190 (numbers and percentages), lines 208-211 (numbers and statistics), lines 217-219 (numbers), lines 316-318 (numbers), lines 320-322 (numbers), lines 324-325 (numbers),

Changes concerning “novelty”:

“a breakthrough in sperm morphology assessment” is now “To summarize, we believe that NMAS will bring as much to the study of morphological defects 338 of the head as CASA has brought to the study of defects of sperm motility.” (lines 348-350)

“Strong broadening” is now “We found for *Cfap43^-/-^* and *Cfap44^-/-^* mice that sperm heads predominantly displayed a "pepper” shape – characterized by a broadening of the base (median widths of body were 3.44±0.09 µM and 3.93±0.09 µM for *Cfap43^-/-^* and *Cfap44^-/-^* respectively, versus 2.95±0.06 µM and 3.09±0.12 µM for *Armc2^-/-^* and *ccdc146^-/-^* bases respectively), …” (starting line 314)

“Such a strong impact on head morphology” (line 322-first submission) sentence has been removed.

2. To improve the audience's readability and interpretation, videos showing the whole images of sperm morphology and motility of the lines harboring multiple heterozygous mutations will be required. For example, the readers will want to see the morphology and motility of representative sperm cells of1het, 2het, 3 het and 4 het.

We agree with the reviewer’s suggestion and now provide representative images of spermatozoa of individuals bearing one to four mutations as Appendix 1-Figure 5. We also provide representative videos of moving spermatozoa as Videos 1 to 5. Respective legends were added in the manuscript.

3. Although the suggested concept is appealing, the current dataset shows that heterozygous mutations on four MMAF-causing genes, CFAP43, CFAP44, ARMC2, and CCDC146, do not result in male fertility issues. Thus, it is not clear how critical oligogenic mutations are in male fertility.

We agree with the reviewer that it is a key point. However, it is well recognized that the mouse model has proved limitations to decipher the function of the proteins involved in sperm physiology and to assess the impact of their lack on fertilization. These limitations are the housing conditions, the mating protocol and the very high fertility of this model.

For instance, concerning the housing conditions, the function of MAGE cancer testis antigens to protect the male germline is revealed only when males are subjected to an environmental stress (Fon Tacer *et al.*, 2019^doi:10.1126/sciadv.aav4832^). For mating protocol, the phenotyping is performed in particular conditions where mutated males are mated with wild-type females, without competition and in breeding conditions masking complex phenotypes. An example of such a limitation of classical reproductive phenotyping has been emphasized in the study on the importance of the Pkdrej protein in sperm capacitation (Sutton *et al.*, 2008^doi:10.1073/pnas.0800603105^). Another remarkable exemple is the phenotype of Enkurin KO mice, with no obvious impact on the litter size, despite the function of the protein in flagellum beating (Jungnickel et al., 2018 ^doi:10.1093/biolre/ioy105^). Overall, the highly significant increase of sperm defects (decrease of sperm motility and increase of head morphological defects) induced by the accumulation of deficient proteins shown in this manuscript would probably impact male fertility in more challenging conditions in mouse.

Moreover, the mouse model has much higher fertility than human. Human spermatogenesis is clearly less efficient than that of mice and the level of male infertility is around 15% whereas is less than 1% in mice. It is therefore to be expected that the various defects we noticed in this study will impact more severely human sperm fertilizing competence.

We have added a paragraph addressing this point (lines 255-273)

In addition, the authors do not present data to explain why accumulated mutations on MMAF causing genes causes accumulative head morphology defects but not flagellar morphology.

Sperm head abnormalities in MMAF individuals, including at the heterozygous state, are not unexpected, and have been reported in the original papers of genes studied (Coutton et al., 2018 ^doi:10.1038/s41467-017-02792-7^; Coutton et al., 2019 ^doi:10.1016/j.ajhg.2018.12.013^) and elsewhere (Hwang et al., 2021 ^doi:10.3389/fcell.2021.662903^). The involvement of “MMAF” proteins in head formation has also been suggested in the original papers of the genes studied. We now discuss extensively the reasons why MMAF genes affect also head formation (see discussion lines 286-300).

It is the area that need to be addressed with more experimental evidence.For example, head morphology defects was shown in Danh2-KO sperm (PMID: ) in which the protein levels or localizations of other axonemal components are demonstrated to be altered. While it is a widely accepted idea that the cytoplasm (mRNAs and proteins) is shared during spermatogenesis by forming syncytium (thus the protein levels of haploid spermatids of heterozygous males are typically in the same levels as those from wild type animals), demonstrating the protein levels and/or localization of CFAP43 CFAP44, ARMC2, and CCDC146 in the sperm bearing these multiple heterozygous mutations will be critical.

Our article does not question the sharing of proteins between the different cells of the syncytium, but rather indicates that the decrease in the expression of different proteins involved in the same cellular function leads to a loss of function, probably due to an instability of molecular interactions. This phenomenon of oligogenism is also observed in diploid cells and is responsible for genetic diseases. Nevertheless, we agree that the suggested experiments on protein localisations would bring very interesting knowledge but unfortunately, we were forced to eliminate our mouse lines due to the covid crisis and repeated lockdowns. Recreate mice with four mutations would take us several years and would require a financial and human investment that we no longer have.

4. From the comparison of +/- and WT sperm head shape in each mouse model, the authors represent that head morphology of CFAP43 and CFAP44+/- sperm are significantly abnormal than WT sperm (Figure 1-2, B). And the proportion of head abnormal sperm between ARMC2+/- males and WT males are not significantly different (Figure 3B). However, the authors' NMAS analyses show that the shapes (angle at each points) of CFAP43+/- and CFAP44+/- sperm head are very similar to those of WT sperm (Figure 1-2, C and D). Yet, ARMC2 +/- sperm show clear difference compared to WT sperm (Figure 3C and D). This discrepancy should be explained clearly.

We fully agree that the comparative evaluation of the contributions of the NMAS computer program compared to those of visual observation is not obvious and deserves a better description, in order to avoid misinterpretations. To this end, we have revised the discussion regarding manual and automated analyses. We added the following section (lines 338-350):

“It should be noted that NMAS analysis is complementary to the human visual analysis made from optical microscopy and does not replace it because the nature of the detected defects is slightly different. Indeed, the software can detect fine nuclear shape anomalies that are undetectable to the human eye and thus reveals morphological variations that are not retained visually by the experimenter. In this study, NMAS revealed a head morphology variation of Armc2^+/-^ versus Armc2^+/+^ sperm (Figure 3B) that had not been detected by visual analysis (Figure 3A). On the other hand, NMAS does not identify some abnormalities evidenced by light microscopy stains such as acrosome or flagellum insertion defects. For instance, this was illustrated by the similarity of the nuclear profiles observed between Cfap43^+/-^, Cfap44^+/-^ and their wild-type counterparts (Figures 1B and 2B), while abnormalities were clearly detected by light microscopy analysis (Figures 1A and 2A).”

5. Lines 241-243: Although the authors represent differences on sperm proportion with head defects, it is quite unclear whether that head shape difference can be considered to "sperm defects" to cause male infertility.

This point has been addressed in essential revision point #4 and in your major comments #3.

In addition, authors should clearly demonstrate why sperm head shape is more affected by the accumulated "MMAF-causing" gene mutation as it will cast the question what is the 'definition' of MMAF.

For three in 4 genes involved in this study, the presence of head defects in MMAF syndrome was already described and the impacts of MMAF genes on head morphology were addressed:

– Concerning CFAP43/44, both proteins are involved in intra-manchette transport (Yu et al., 2021 ^doi:10.1017/S0967199420000556^), a key organelle in head shaping. It has indeed been shown that sperm deficient for manchette associated proteins such as Hook1 (Mendoza-Lujambio et al., 2002 ^doi:10.1093/hmg/11.14.1647^), Clip170 (Akhmanova et al., 2005 ^doi:10.1101/gad.344505^), Rim-bp3 (Zhou et al., 2009 ^doi:10.1242/dev.030858^), or Azil (Hall et al., 2013 ^doi:10.1371/journal.pgen.1003928^), exhibit abnormal head morphology.

– Regarding ARMC2, its implication in intraflagellar transport has recently been shown (Lechtreck et al., 2022 ^doi:10.7554/*eLife*.74993^) and several previous reports associated sperm head malformations with IFT genes, like IFT20 (Zhang et al., 2016 ^doi:10.1091/mbc.E16-05-0318^), IFT25 (Liu et al., 2017 ^doi:10.1093/biolre/iox029^), IFT27 (Zhang et al., 2017 ^doi:10.1016/j.ydbio.2017.09.023^), or IFT88 (San Agustin et al., 2015 ^doi:10.1091/mbc.E15-08-0578^).

– Finally, we cannot say much about the Ccdc146 protein because the full description of the KO is submitted elsewhere, but Ccdc146 is located in the centriole, an organelle for which defects are associated with head anomalies, and in particular in the flagellum attachment zone (Lv et al., 2020^doi:10.1136/jmedgenet-2019-106479^).

These scientific points to understand why head defects are observed for MMAF genes in this study are now clearly discussed (lines 286-300).

Concerning the definition of the MMAF syndrome, we agree that the definition of MMAF is incomplete. However, not all MMAF genes induce morphological abnormalities of the head, FSIP2 for example (Martinez et al., 2018^doi:10.1093/humrep/dey264^) and maybe the MMAF syndrome could be split in pure MMAF and MMAF and the head (MMAFH). However this consideration is out of the scope of this manuscript.

The authors interprets that their observation that accumulation of heterozygous mutations leads to marked head defects without obvious morphological flagellar defects suggest 'head shape is more prone to collapse than flagellum biogenesis'. This is highly speculative – those MMAF gene products could simply participate in head morphogenesis before targeting, and/or the less efficiently transporting/targeting of the products simply could remain in the cell body, thus interfereing streamlined shaping of the head (as suggested in 260-262).

We fully agree that this sentence was too speculative. It was removed and we now present instead different hypothesis as regard of the function of the different genes as presented in the previous comment (lines 286-300).

Just as the comment for the point #3, demonstrating the protein levels and/or localization of CFAP43 CFAP44, ARMC2, and CCDC146 in the sperm bearing these multiple heterozygous mutations will be critical.

As for your major comment number 3, we fully agree that these experiments would bring very interesting knowledge but as our mouse lines had to be eliminated following the covid crisis and repeated lockdowns, we do not possess the required biological resources to perform these experiment anymore.

6. In general, it is suggested for the authors to clarify the claims/conclusion supported by their data vs. speculation throughout the text.

Following this and previous comments, we have added numerical data and made several changes to the text of the manuscript to remove subjectivity.